# Position: Falsify, Don't Just Discover - AI-Generated Discoveries are NOT Born Scientific

## Abstract

Rapid development of artificial intelligence has drastically accelerated the development of scientific discovery. Recently, the rise of Large Language Models (LLMs) has led to the prosperity of autonomous agents, which enable scientists to seek references at different stages of their research. The demonstrated autonomy of these agents has led to designations such as "AI Scientist". However, it remains an open question whether we have truly reached the stage where scientific discovery can be fully automated. In this paper, *we posit that automated scientific discovery needs automated falsification, a missing part in the current research*. As stated in Popper (1935), the central component of scientific research is falsification, where experiments are designed and executed to validate or refute hypotheses. To automate scientific discovery, the falsification process should also be automated. We review the substance of falsification in each stage along the development of AI-accelerated scientific discovery, and analyze the subject, the object, and the degree of automation of the falsification process. Following this, we initiate BABY-AIGS, a proof-of-concept AI-generated discovery system enabled by automated falsification. Through qualitative and quantitative studies, we reveal the feasibility of automated falsification, and advocate for responsible and ethical development of such systems for research automation.

## 1. Introduction

Deep learning has revolutionized scientific research (LeCun et al., 2015; Vaswani et al., 2017; Jumper et al., 2021; Achiam et al., 2023). Leveraging the enormous amount of experimental data, deep learning methods extract the underlying patterns in an end-to-end manner and effectively generalize to unobserved scenarios. The breakthroughs from deep learning in scientific domains, such as protein structure prediction (Jumper et al., 2021), gravitational wave detec-

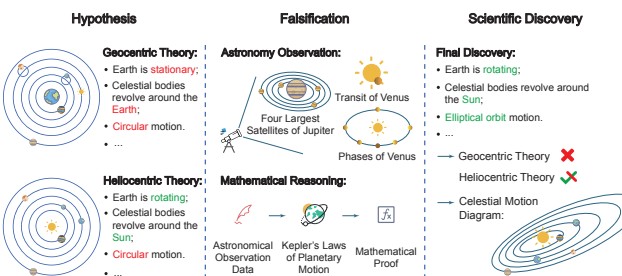

*Figure 1.* Examples of scientific research processes conducted by human researchers. Explicit falsification serves as a vital stage to falsify or verify the proposed hypotheses from either empirical or theoretical experiments, leading to the ultimate scientific discovery.

tion (George & Huerta, 2018), and plasma control (Degrave et al., 2022), have received award-winning recognition. As a result, AI for Science has emerged as a highly-regarded research field (Wang et al., 2023a).

In the paradigm of AI for Science, AI primarily serves as a tool to assist researchers in making discoveries. With the rapid development of foundation models and autonomous agents (Park et al., 2023), AI techniques nowadays boast the capabilities of general-purposed textual understanding and autonomous interaction with the external world. These capabilities lead to the successful applications of AI-as-research-assistants, ranging from single-cell analysis (Hou & Ji, 2024) to drug discovery (Wang et al., 2023b). The capability of providing research assistance leads to a more ambitious challenge: *Can foundation model-powered agents be autonomous researchers, independently completing the entire process of scientific discovery, thereby **transforming AI for Science into AI-Generated Science (AIGS)**?*

When constructing an AIGS system with full process autonomy, the design criteria of the system should refer to the definition of the scientific research process itself. As stated by Popper (1935), scientific research follows a systematic process of proposing novel hypotheses, conducting experiments through trial and error, and falsifying these hypotheses to conclude (refer to Figure 1 for an example). Although creativity is widely believed to be indispensable in the research process - which is also accounted for in previous work (Si et al., 2024) - the central component of scientific re-

search is *falsification*: designing and executing experiments to validate or refute hypotheses, and falsified hypotheses also pose positive contributions to scientific progress (Lowe et al., 2025). In practice, experienced researchers accumulate practical skills or reusable workflows (Gil et al., 2007) from hands-on experimentation, which eases the experimental design and execution process. The purpose of conducting experiments is still to falsify their proposed hypotheses.

In this paper, we posit that *AI-Generated Science* **should also undergo the automated falsification process**. To elaborate, automated falsification is to proactively and autonomously seek feedback from experimental environments to refute research proposals: Automated falsification is to AI-empowered scientific discovery, as Popper's definition of falsification is to human-empowered research process. Following this principle of automated falsification, the handful of recent works related to AI-empowered scientific research deserve a cautious re-examination, which can be roughly divided into three lines:

- In the first line, researchers propose performance-optimizing frameworks with or without agentic abilities to maximize the utility of an automated machine learning system (Jumper et al., 2021; Juan et al., 2021; Alfarano et al., 2024). While these systems have achieved remarkable progress in specific domains, they are refrained from AIGS because of a lack of automated falsification for the research progress: Their objectives to be optimized have been hard-coded in the system; however, the falsification objective should have been the dedicated systems themselves.

- In the second line, several efforts have been made to leverage the capability of frontier models to discuss the possibility of proposing scientific hypotheses. For instance, Si et al. (2024) investigate the ability of LLMs to generate research ideas on language model prompting, while Ding et al. (2024) study the autonomy of proteomics research proposals. This line of research is limited in that only the prologue parts are emphasized in terms of AI-powered automation.

- The third line of research aims to design systems that automate the full process of scientific discovery. MLR-copilot (Li et al., 2024b) takes existing research papers as input and produces execution results by both generating ideas and implementing experiments. AI Scientist (Lu et al., 2024) propose to organize the ideas and experimental results into research papers as the output. This line of research arouses significant excitement in the community, but is feedbacked with controversy: Criticisms include the incremental nature of the knowledge generated "tweaks", as well as the poor quality of the generated code and the presentation of the paper (Koppel, 2025). In fact, as further benchmarked by DiscoveryWorld (Jansen et al., 2024), Discovery-

Bench (Majumder et al., 2024b), DSBench (Jing et al., 2024), and ScienceAgentBench (Chen et al., 2024d), an automatic AIGS system that produces novel research from beginning to end is still in the early stages, and significant gaps remain underexplored, especially in the area of *autonomous falsification*.

In this work, we initiate BABY-AIGS, our baby-step attempt toward a full-process AIGS system. BABY-AIGS comprises several LLM-powered agents, each responsible for distinct stages within the research workflow, mimicking the full-process human research that falsifies hypotheses based on empirical or theoretical results for scientific discoveries. BABY-AIGS operates in two phases: the first phase iteratively refines proposed ideas and methods through enriched feedback, incorporating experimental outcomes, detailed reviews, and relevant literature. The second phase emphasizes explicit *falsification*, a key feature absent in prior systems (Lu et al., 2024), executed by FALSIFICATIONA-GENT. Based on experimental results related to the proposed methodology, the agent identifies critical factors likely contributing to notable experimental phenomena, formulates hypotheses, and ultimately produces scientific discoveries verified through ablation experiments. We apply BABY-AIGS across three open-world research topics: *data engineering*, *self-instruct alignment*, and *language modeling*, and a closed-world environment, *DiscoverWorld++*, which is modified from Jansen et al. (2024) with a focus on evaluating falsification capability of agentic frameworks. Empirical results indicate that BABY-AIGS can autonomously produce meaningful scientific discoveries from automated falsification, supported by qualitative analysis, demonstrating the feasibility and necessity of this approach. Nevertheless, the performance of BABY-AIGS still lags behind that of experienced researchers in top academic venues, suggesting avenues for further enhancement. Our proof-of-concept studies shed light on the pros and cons of the further development of such AI-generated science. We believe that automating falsification is vital for developing AIGS systems towards automating rigor and solid scientific discoveries.

## 2. The Development of AI-Accelerated Scientific Discovery

In this section, we review and envision the development of AI-accelerated scientific discovery as four paradigms (Figure 2): (I) **AI as a Performance Optimizer**, where deep neural networks are trained with large-scale observation data in a specific scientific problem to extract the patterns in an end-to-end manner. In this paradigm, the AI techniques are used to optimize the specific prediction / regression performance in the pre-defined scientific problem with the consideration of out-of-domain generalization. (II) **AI as a Research Assistant**, where LLM-driven research copilots

*Figure 2.* Overview of the four paradigms of AI-accelerate scientific discovery systems.

are used to assist the human research process. The synergy between Paradigm (I) and (II) forms the AI-powered acceleration of scientific discovery nowadays. (III) **AI as an Automated Scientist**. In this regime, foundation model empowered agents with scientist-like behavior should complete the entire research process, ranging from the initial idea proposal to the ultimate delivery of the scientific findings. (IV) **AI Forms a Research Community**. Upon the prosperity of fully-autonomous AI researchers depicted in the previous stage, we envision the collaborations among the agentic researchers foster an AI-formed research community.

### 2.1. AI as a Performance Optimizer: Discoveries in Specific Tasks

With the rise of deep learning, AI has significantly impacted scientific discoveries across various fields, particularly in optimizing specific tasks by exploring well-defined search spaces or extracting patterns from piles of data. Utilizing specialized deep learning models, scientific breakthroughs continue to emerge across diverse fields, including accurate protein structure prediction (Jumper et al., 2021; Abramson et al., 2024), drug discovery and materials design (Gilmer et al., 2017; Juan et al., 2021), and the simulation of physical systems (Sanchez-Gonzalez et al., 2020). Moreover, a long-standing open problem in mathematics has been resolved by training a specialized Transformer-based expert (Alfarano et al., 2024). Deep learning models are widely recognized to be highly effective in learning representations and patterns from data to assist the development of scientific discovery.

Large Language Models (LLMs), equipped with extensive world knowledge and advanced reasoning, have empowered increasingly creative and autonomous agents. They have demonstrated remarkable proficiency in autonomously developing evolutionary strategies for instruction datasets (Zeng et al., 2024), identifying and rectifying their weaknesses (Cheng et al., 2024; McAleese et al., 2024),

and optimizing organizational structures for improved efficiency (Zhang et al., 2024a; Hu et al., 2024a), highlighting their potential for performance optimization through structured search. Beyond language tasks, their creativity contributes to impressive discoveries in scientific fields. Via scientifically oriented, logically organized searches, LLMs can be guided to discover mathematical solutions (Romera-Paredes et al., 2024) and physical equations (Ma et al., 2024; Shojaee et al., 2024). Augmented with specialized tools and verification engine, LLMs are capable of solving advanced geometry problems (Trinh et al., 2024), designing chemical reactions (Chen et al., 2024a) and discovering novel materials (M. Bran et al., 2024; Ghafarollahi & Buehler, 2024). The limitation of this line of research is that the falsification process does not exist for either the training or the inference phase of the AI model. It is the objective to be optimized that needs falsification; The objective, however, is hard-coded as the loss function for the model in advance.

### 2.2. AI as a Research Assistant: Copilots in Human-AI Collaboration

Equipped with expanding scientific knowledge and generative capabilities, LLMs gradually exhibit great potential to assist researchers at various stages of the research process.

**Literature review** is a critical yet tedious step for scientific research, prompting the use of autonomous LLM-based solutions. Advanced LLMs can identify relevant sources, generate structured summaries, and organize studies into hierarchical structures (Haman & Školník, 2024; Huang & Tan, 2023; Sharma et al., 2021; Li et al., 2024c). Retrieval-augmented frameworks also help produce more reliable and comparative literature reviews (Hsu et al., 2024). Overall, LLM-based agents have demonstrated the capability to generate readable and detailed overviews of existing research.

For **research ideation**, LLMs can generate reasonable hy-

potheses based on internal knowledge and additional inputs. While a large-scale human study (Si et al., 2024) finds these ideas to be more novel but less feasible than those from experts, other evaluations (Kumar et al., 2024; Girotra et al., 2023) also recognize the potential of LLMs as sources of inspiration. Recent multi-agent frameworks based on scientific literature aim to accelerate research proposals (Baek et al., 2024; Nigam et al., 2024a;b), yet balancing novelty and feasibility remains challenging (Si et al., 2024), and evolving initial proposals into validated knowledge still demands substantial falsification effort.

AI-assisted **idea implementation and automatic experimentation** often take place at the repository level, and take advantage of the increasingly stronger coding capabilities of LLMs. Several benchmarks (Jimenez et al., 2024; Liu et al., 2023; Chan et al., 2024) target machine learning and software engineering tasks, while others (Yang et al., 2024a; Wang et al., 2024; Tao et al., 2024) leverage agentic collaboration to reduce the coding workloads of researchers. However, fully autonomous end-to-end experimentation imposes higher demands on coding agents. Challenges, such as low success rates (Lu et al., 2024) and misalignment between ideas and implementations, underscore the need for better reliability and enhanced automatic falsification.

In the realm of **academic writing**, LLMs can be utilized for drafting structured outlines, refining human-written texts and presenting research findings. Recent studies (Liang et al., 2024b; Geng & Trotta, 2024) have demonstrated a steady increase for LLM usage in scientific writing. This trend presents both opportunities and challenges for academia. When properly used, LLMs could improve research efficiency and presentation; But when misused, risks emerge as well in terms of research integrity. Therefore, effective oversight through detection strategies (Liang et al., 2024a; Yang et al., 2024b; Ghosal et al., 2023) and watermarking techniques (Kirchenbauer et al., 2023; Zhao et al., 2023; Zhang et al., 2024b) is both beneficial and necessary.

Additionally, following LLM-as-judge methods (Zheng et al., 2023), LLM-based agents are employed for comprehensive **evaluation** on research outputs (Lu et al., 2024; Li et al., 2024b). Comparing model-generated reviews with expert evaluations, researchers have evaluated the capabilities of LLMs to provide insightful and high-quality reviews by constructing meticulously annotated datasets (Du et al., 2024) or training preference models (Tyser et al., 2024). With multi-agent collaboration to promote in-depth analysis and constructive feedback, D'Arcy et al. (2024), Jin et al. (2024) and Yu et al. (2024) develop LLM-powered agent pipelines to perform paper reviews, helping researchers improve the quality of their papers. Furthermore, Sun et al. (2024) introduces a reviewing tool designed to support reviewers with knowledge-intensive annotations. In a no-

table development, ICLR conference adopt reviewer agents to provide constructive feedback on human-submitted reviews, showcasing a promising application of AI-assisted reviewing (ICLR Blog, 2024). Recently, researchers also constructed benchmarks for AI as a research assistant at more than one stage above (Lou et al., 2024). While LLMs offer reliable research feedback, current approaches only automate select parts of the research workflow and largely neglect the falsification process.

## 2.3. AI as an Automated Scientist: Towards End-to-end Scientific Discovery

Structured in well-organized agentic pipelines, LLMs are increasingly capable of tackling complex tasks collaboratively, with end-to-end scientific research being one of the most ambitious and challenging applications. For instance, Lu et al. (2024) develops an iterative multi-agent framework that supports the entire research process, from proposing novel ideas to presenting polished findings. Similarly, Li et al. (2024b) introduces an automated research system for machine learning, and Manning et al. (2024) employs LLMs to simulate scientists for social science research. Beyond research systems, Jansen et al. (2024) proposes a simulation environment designed to challenge agents in automated scientific discovery. Despite these advancements, current end-to-end research systems still fall short of generating falsifiable scientific findings, constrained by the capabilities of both designed framework and foundation models. While previous research (Lu et al., 2024) has yielded well-formulated outcomes, the vision of automated science discovery still requires further efforts, as the efficacy of existing systems still falls short of human researchers in terms of falsification process: validate or refute the proposals they made by elaborate experiment design and execution.

## 2.4. AI Forms a Research Community: Enable Academic Swarm Intelligence

Collaborations and debates among researchers have historically driven scientific progress onward. We envision that a community of agentic scientists could greatly accelerate automated discovery. By orchestrating LLM-driven agents to exhibit human-like trustworthy behaviors and assume assigned roles (Park et al., 2022; Gao et al., 2024; Park et al., 2023; Li et al., 2024a; Hua et al., 2023; Xu et al., 2023), early-stage simulations of research communities already show promise for fully automated, AI-driven research.

## 3. Automating Science through AI-Empowered Falsification

In this section, we elaborate falsification is the essence that separates scientific discoveries from random ones, and AI-empowered falsification could automate the AIGS process.

*Table 1.* Comparative analysis on representative AI-powered scientific discovery works and ideal AIGS systems. Works without explicit falsification process could perform falsification implicitly by chance. Automated scientific discoveries require minimal human intervention. "Generalized" indicates whether the work generates scientific discoveries extended beyond the target domain.

| Representative Work | Target Domain | Falsification Process | | | Scientific Discoveries | |
|---|---|---|---|---|---|---|
| | | Explicit | Conductor | Approach | Automated | Generalized |
| AlphaFold (Abramson et al., 2024) | Protein | ✗ | Human Researchers | Wet Experiments | ✗ | ✗ |
| Laboratory Mobile Robots (Dai et al., 2024) | Synthetic Chemistry | ✓ | AI-Empowered Robots | Synthesis & Screening with Physical Equipment | ✗ | ✗ |
| AI Scientist (Lu et al., 2024) | Machine Learning | ✗ | AI-Empowered Agents | Experimenting Through Codebase Edits | ✗ | ✗ |
| AI Hilbert (Cory-Wright et al., 2024) | Polynomial Data | ✗ | AI-Empowered Agents | Polynomial Optimization | ✗ | ✗ |
| DataVoyager (Majumder et al., 2024a) | Data Analysis | ✓ | AI-Empowered Agents | Tool Learning with LLMs | ✓ | ✗ |
| *Ideal AIGS Systems* | *General* | ✓ | *AI-Empowered Systems* | *Experiments Designed & Conducted Autonomously* | ✓ | ✓ |

## 3.1. Scientific Discoveries in Human Research

The typical research process for human scientists (Chen, 2009; Popper, 1935) could be summarize into two main stages: the **pre-falsification** stage, which encompasses exploration of research ideas, refinement of methodologies, and theoretical or empirical analysis, and the **falsification** stage, which involves deriving hypotheses about scientific laws and validating these hypotheses based on theoretical or empirical findings. The objective is to generate scientific discoveries thereby contributing to the human knowledge. As shown in Figure 1, random discoveries could be in the form of false positives or false negatives, which are not scientifically meaningful, e.g., the geocentric theory; in contrast, scientific discoveries are falsifiable so that they could be validated through rigorous experimentation or theoretical analysis, e.g., contemporary celestial mechanics falsified by astronomy observations and mathematical calculations, representing the principles of natural world.

A scientific discovery is a hypothesis that has been rigorously tested and validated through empirical or theoretical means, demonstrating its consistency with observed phenomena. Thus, **scientific discoveries are falsifiable and should have undergone falsification based on empirical or theoretical results**, in contrast to unverified discoveries. Formally, the falsification process can be described as:

$$\mathcal{H} \xrightarrow{\mathcal{E}} O \quad \Rightarrow \quad O \neq \mathcal{H} \quad \text{implies } \mathcal{H} \text{ is falsified}, \quad (1)$$

where $\mathcal{H}$ is the set of hypotheses, $O$ represents the available empirical observations or theoretical results, $\mathcal{E}$ denotes experimental, observational, or reasoning processes, and $O \neq \mathcal{H}$ here represents the contradiction between the empirical observations or theoretical reasoning and the predictions of the hypotheses $\mathcal{H}$. Based on the definition, we also term $\mathcal{H}$ as candidates for scientific discoveries.

In research fields like machine learning, falsification process is performed empirically, i.e. ablation studies, which are collected after researchers design the methodology, build a system, and conduct experiments. Other fields operate differently. For example, in physics or biology, empirical results are observed or gathered from equipment after the experimental design and execution, while in mathematics or the humanities, theoretical insights are often derived through logical reasoning or literature review rather than empirical experimentation. Then, those empirical and theoretical results are used to falsify hypotheses proposed ahead. These root falsification processes of different subjects in distinct knowledge source. In this work, we primarily focus on empirical subjects that requires actual implementation of the methodology to obtain empirical results for falsification, e.g., machine learning, and leave other venues for future work.

## 3.2. AIGS from AI-Empowered Falsification

Human scientific research workflow above reflects the design principles of a full-process AIGS system, which could autonomously take the topic of a research field, an accessible and configurable experiment environment, and other optional resources like a literature base as the input, and output a verbal scientific discovery and the falsification process that support or falsify it.

As depicted in Section 3.1, pre-falsification phase is an indispensable part of the autonomous research process. This phase could contain several stages, such as *idea formation*, *methodology design*, *experiment execution*, *result analysis*, etc., aiming to explore and refine the proposed idea and methodology through feedback including experimental outcomes, reviews based on literature or inherent knowledge in the system, etc. Though highest benchmarking results are typically used as the evidence to show the effectiveness of the methodology and often reflects meaningful scientific

discoveries, the hidden scientific law may also lay in the methods that cause abrupt decreased performance or even subtle changes on a handful of data samples. Those observations should be all recorded in the experimental results as significant phenomena used for falsification.

The core of the system is the automated *falsification* process, which is the foundation to automate scientific discoveries. As depicted above, it starts after the pre-falsification stage, when the methodology is developed through refinement and experimentation. The falsification process is fundamentally established upon designing and conducting ablation studies to verify any key factors that contribute to significant experimental phenomenon. Ideally, given the research history of methodology, the AIGS system could identify all potential candidates for scientific discoveries, i.e. hypotheses, and conduct rigorous ablation experiments attempting to falsify them. Finally, candidates that survive the falsification process are considered as verified scientific discoveries.

Formally, the process could be described as:

$$Scientific\ Discovery = \text{Falsification}\left(Research\ History\right),$$
$$(2)$$

where $\text{Falsification}(\cdot)$ represents the AI-empowered workflow of the falsification process, and

$$Research\ History = \left\{Method.^{(i)}, Exp.\ Results^{(i)}\right\}_{i=1}^{M},$$

where $M$ is the number of experiments, $Method.^{(i)}$ is the methodology of the $i$-th experiment, and $Exp.\ Results^{(i)}$ is the empirical results of the $i$-th experiment.

However, there are a few challenges in automating falsification process: (1) **Identifying the experimental phenomena most likely to yield meaningful scientific discoveries**. Meaningful scientific findings are often scarce during the initial stages of methodological exploration. The changes in the empirical results between experiments may serve as an indicator. (2) **Designing and performing valid ablation experiments**. A gap frequently exists between the adjustable parameters in an experiment and the factors intended for ablation. Furthermore, when the variance of empirical results is substantial, multiple experimental iterations are necessary to draw reliable conclusions. (3) **Evaluating and validating hypotheses based on experimental outcomes**. Even when ablation experiments are successfully executed, the resulting data may not conclusively confirm the validity of the hypothesis, as other potential confounding factors could influence the interpretation of the results. To practically implement the automated falsification process, the system should be designed with the following principles in mind:

- To achieve smooth and consistent experimentation, we emphasize the importance of *executability* of the proposed methodology, which serves as the basis for collecting empirical results for both method development and automatic *falsification*.

- The *creativity* of the generated proposals from the AIGS system is the overall objective of the research. However, the creativity becomes authentic only when accompanied by rigorous results from *falsification*.

To sum up, *falsification* is the foundation of a full-process AIGS system, pillared by experimenting scaffolds accounting for *executability* and targeting at the ultimate goal of high research *creativity*.

## 4. Alternative Views

In this section, we explain why current works in AI-accelerated scientific discovery fail to achieve full-process automated AIGS systems, as shown in Table 1.

**Could evolving performance optimizers in specific domains automate science?** Evolving performance optimizers, such as those built with reinforcement learning (OpenAI, 2024; DeepSeek-AI, 2025; Kimi et al., 2025) and self-improving algorithms (Ding et al., 2024), could automate certain aspects of research process with enhanced reasoning capabilities or improved workflows. However, without experimentation in the loop and access to massive literature, these systems refrain from understanding the hidden laws of nature, which is the ultimate goal of scientific discovery. This aligns with the observation that systems such as AlphaFold (Abramson et al., 2024) still require substantial human-conducted experimental verification for validation.

**Could AI-empowered ideation and experimentation automate science?** AI-empowered ideation and experimentation are important for research process. This approach is valid when the objective is the outcome from the experiment or the methodology itself rather than scientific discoveries. For instance, Laboratory Mobile Robots (Dai et al., 2024) could achieve autonomous synthesis workflows without insightful hypotheses. In contrast, AI Scientist (Lu et al., 2024) could generate hypotheses anyway but falsify them largely by chance. The missing explicit falsification process leads to failures in streamlining scientific discoveries.

**Could AI-empowered data interpreters automate science?** Data-driven discovery hinges on two key assumptions: first, that hypotheses generated by AI are falsifiable, and second, that discoveries are constrained by the scope of the available dataset. Thus, falsification could be implemented with limited verification, such as symbolic (e.g., AI Hilbert (Cory-Wright et al., 2024)) or statistical (e.g., DataVoyager (Majumder et al., 2024a)) methods. However, the limited scope of the dataset can introduce biases during data manipulation, which can distort findings. These systems are highly specialized and domain-specific, inhibiting broader application without human intervention.

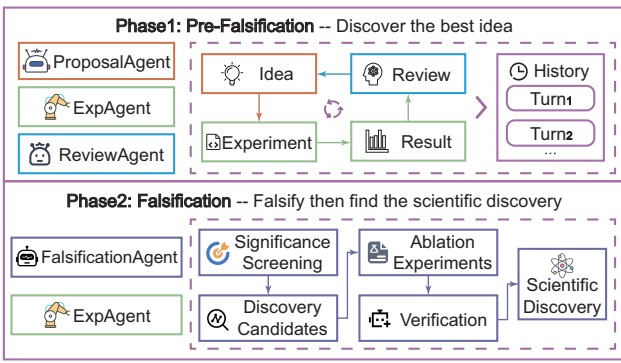

*Figure 3.* Overview of our BABY-AIGS system design. The upper part denotes **Pre-Falsification** phase iteratively discovering research ideas for methodology design and implementation. This process summons multi-turn logs as the history context, based on which FALSIFICATIONAGENT could produce scientific discoveries during the **Falsification** phase, as shown in the lower part.

## 5. Proof-of-Concept Experiments with BABY-AIGS System

Following the principles in Section 3.2, we elaborate on a baby-step system towards the full-process AIGS in this section, including system design, experiment setup and results.

### 5.1. BABY-AIGS System Design

The BABY-AIGS system is an LLM-powered multi-agent framework for automated scientific discovery, including PROPOSALAGENT, EXPAGENT, REVIEWAGENT, FALSIFICATIONAGENT, and other optional agents. It operates in two phases (Figure 3): **Pre-Falsification** and **Falsification**. In the **Pre-Falsification** phase, the system iteratively refines research ideas and methodologies through stages like idea formation, experiment execution, and result analysis, using feedback to improve hypotheses. The **Falsification** phase then conducts ablation studies to systematically test and verify key factors contributing to significant experimental phenomena, identifying validated scientific discoveries. Additionally, the system incorporates a Domain-Specific Language (DSL) (Mernik et al., 2005) and multiple-sampling strategies to enhance efficiency and executability. A detailed system design is provided in Appendix B.

### 5.2. Experiment Setup

We conduct experiments and analysis on two types of research topics: *open-world topics* and *closed-world environments*. The open-world research topics include *Data Engineering*, *Self-Instruct Alignment*, and *Language Modeling*, all focusing on fundamental challenges in machine learning research. The closed-world environment, *DiscoveryWorld++*, evaluating scientific discovery abilities within a self-contained environment, emphasizing the process of

falsification. Detailed descriptions of these tasks and the implementation of the BABY-AIGS system are provided in Appendix C.1 and Appendix C.2.

### 5.3. Evaluation of Falsification

We assess the ability of BABY-AIGS to perform falsification through human evaluation, focusing on the falsification process carried out by FALSIFICATIONAGENT. This process involves hypothesizing potential influencing factors, identifying the key variables that may impact experimental results, designing and conducting ablation experiments, and ultimately validating the real factors contributing to the experimental significance. The human evaluation is carried out by volunteer researchers with experience in publishing at top-tier conferences. Evaluators assess the falsification process based on three key dimensions, each scored on a scale from 0 to 2 (higher is better):

- *Importance Score*: This score reflects the importance of the scientific discovery candidate. It evaluates the extent to which the identified factors can influence the experimental results, considering their relevance and potential impact with the primary experiments.

- *Consistency Score*: This score assesses whether the proposed ablation experiment plan is aligned with the identified scientific discovery candidate. It considers whether the experiments are designed to ablate the factor of interest and appropriately test the hypothesis.

- *Correctness Score*: This score evaluates the accuracy of the final scientific discovery derived from the ablation studies. It considers whether the conclusions drawn from the results from ablation experiments are correct, based on the observed empirical results.

- *Overall Score*: This score is the average of all other dimensions mentioned above for each sample, serving as a comprehensive indicator of the quality of the falsification process.

Additionally, several studies from the top conferences (Liu et al., 2024a; Chen et al., 2024b; Li et al., 2024d; Zhao et al., 2024) are included in the evaluation set to serve as a baseline. We conduct the evaluation on our three open-world research topics, with statistic results shown in Table 2, where the p-values are obtained from a left-tailed Welch's t-test on 60 samples against the top conference baseline and the gap is considered significant when $p < 0.05$. The guidelines given to evaluators for the human evaluation are detailed in Appendix C.3, while the results for individual research topics and specific cases are presented in Appendix D.4 and Appendix D.5, respectively. From these results and cases we have the following conclusions:

*Table 2.* Statistic results of human evaluation on the falsification process in our three open-world research experiments.

| Metric | AVG | STD | P-Value | MIN | MAX |
|---|---|---|---|---|---|
| **Importance Score** ($0 \sim 2$) | | | | | |
| BABY-AIGS (Ours) | 1.72 | 0.49 | 0.00 | 0.00 | **2.00** |
| Top Conference | **2.00** | 0.00 | — | **2.00** | **2.00** |
| **Consistency Score** ($0 \sim 2$) | | | | | |
| BABY-AIGS (Ours) | 1.12 | 0.67 | 0.00 | 0.00 | **2.00** |
| Top Conference | **2.00** | 0.00 | — | **2.00** | **2.00** |
| **Correctness Score** ($0 \sim 2$) | | | | | |
| BABY-AIGS (Ours) | 0.97 | 0.78 | 0.00 | 0.00 | **2.00** |
| Top Conference | **2.00** | 0.00 | — | **2.00** | **2.00** |
| **Overall Score** ($0 \sim 2$) | | | | | |
| BABY-AIGS (Ours) | 1.27 | 0.43 | 0.00 | 0.33 | **2.00** |
| Top Conference | **2.00** | 0.00 | — | **2.00** | **2.00** |

(1) **BABY-AIGS could produce valid scientific discoveries with falsification process.** Table 2 shows the maximum value of each metric is tied to the top-conference baseline, indicating that FALSIFICATIONAGENT could produce valid scientific discoveries in current design. Additionally, the average importance score is notably higher than both the consistency and correctness scores, suggesting that while FALSIFICATIONAGENT can identify key factors potentially relevant to a scientific discovery, it struggles to formulate concrete experimental plans and verify hypotheses. This limitation may stem from the constraints of the foundation model or the lack of high-quality demonstrations of experimental design within the prompts. Looking ahead, as AI continues to advance, automated falsification is expected to become more efficient and accurate, playing an increasingly positive role in the automation of scientific insights.

(2) **BABY-AIGS still lags significantly behind top human researchers.** The p-values in Table 2 suggest that the falsification process of BABY-AIGS is notably less satisfactory compared to the existing literature from top conferences, as evaluated from a human perspective. This underscores the need for further improvements to advance automatic falsification from merely feasible to truly reliable. This finding aligns with our perspective that automatic falsification deserves greater attention in AIGS research.

(3) **AI-generated discoveries are NOT born scientific and falsification is the missing link.** Examples in Appendix D.5 demonstrate that some discoveries made by the system appear promising, as performance improvements have been observed in experiments. However, automatic falsification conducted by FALSIFICATIONAGENT reveals that the identified factors are not causally linked to the experimental results. For instance, in the second example of FALSIFICATIONAGENT (Appendix D.5.1), the system initially identifies *context retention* and *logical progression* as key criteria for performance improvement. However, FALSIFICATIONAGENT conducts an automatic ablation study by removing one criterion and finds that "*the marginal improvements in scores suggest that these principles alone*

*do not significantly enhance data quality*". The analysis concludes: "*The true scientific discovery is that while context retention and logical progression are important for multi-turn dialogues, their isolated application does not dramatically improve the dataset's quality for MT-bench. This suggests the need for a more nuanced and integrated approach, considering other quality metrics alongside these principles.*" This underscores the critical role of falsification in transforming discoveries into scientifically valid ones. In contrast, as shown in Table 1, explicit automatic falsification is largely overlooked in existing works, further emphasizing the importance of falsification in AIGS.

Moreover, **current scientific discovery benchmarks also lack considerations of evaluating falsification performance.** Results on DiscoveryWorld++ show that in some cases, even if the agent succeed in finishing all the sub-tasks and being marked as success according the evaluation original metrics of DiscoveryWorld, however, as shown in Appendix D.5.4, it fails explicitly answering the question regarding the scientific discovery when falsification is necessary in order to reach the right answer. Current scientific discovery benchmarks, represented by DiscoveryWorld, did not involve falsification as an essential part of the task process and led to an incomplete evaluation of research agents.

**5.4. Supplementary Evaluations**

As discussed in Section 3.2, *falsification*, *creativity*, and *executability* are the three desiderata of AIGS systems. Accordingly, we evaluate *creativity* and *executability* with AI Scientist (Lu et al., 2024) as the baseline. Due to space constraints, detailed evaluation results are provided in Appendices C.3, C.4, and D. The results are briefed as follows: (1) BABY-AIGS demonstrates strong *creativity* in research idea exploration and refinement; (2) BABY-AIGS exhibits high *executability* in experimentation and the full research process; (3) BABY-AIGS outperforms AI Scientist in both aspects. We attribute the superiority of BABY-AIGS to the integration of the *falsification* modules.

# 6. Conclusion

While AI-powered systems have shown promise in automating distinct stages of scientific discovery, fully autonomous and reliable AI systems for scientific research require the ability to falsify hypotheses. We posit that AI-empowered scientific discovery must incorporate automated falsification to ensure the validity and credibility of AI-generated discoveries. we highlight the underlying challenges and opportunities for this by analyzing the subject, object, and degree of falsification with several existing AI systems that accelerate scientific discoveries. Looking ahead, the responsible development of AI-generated science must prioritize rigorous falsification mechanisms to prevent unreliable conclusions.

## Impact Statement

In our BABY-AIGS system, the agent does not perform harmful operations in the environment because of the design of DSL, task constraints, and lack of access to external tools. However, while the system developed in this study is limited in scope, AIGS systems as a whole may have significant impacts in the future, with potential risks that should not be overlooked. This section explores the potential negative impacts of such systems, drawing on prior research, and offers suggestions for promoting their positive development.

### A. Potential Negative Impacts of AIGS Systems

**Impact on Human Researchers and Academic Community.** In the absence of robust publication standards and academic review processes, AIGS systems could flood the academic community with low-quality literature, which will further increase researchers' workload and disrupt the efficient dissemination of knowledge (Lu et al., 2024; Si et al., 2024; Hu et al., 2024b). And although Si et al. (2024) and Kumar et al. (2024) suggest that LLMs can generate ideas more creative than humans, the extent of such creativity remains uncertain. LLM-powered AIGS systems tend to rely heavily on existing data and patterns, which could foster *path dependency* and limit opportunities for groundbreaking discoveries. Additionally, these systems might inadvertently use proprietary or copyrighted material, raising concerns about intellectual property infringement (Kumar et al., 2024). Furthermore, AIGS systems also present several unpredictable challenges for human researchers:

- **Dependence Effect and Cognitive Inertia**: Overreliance on AI-generated insights may diminish researchers' independent thinking, leading to cognitive stagnation and a decline in critical thinking skills (Si et al., 2024; Hu et al., 2024b).

- **Ambiguity in Responsibility Attribution**: The involvement of AI complicates the assignment of credit and responsibility, potentially disrupting existing incentive structure (Si et al., 2024; Hu et al., 2024b).

- **Weakened Collaboration and Increased Isolation**: As AIGS systems become capable of independently generating publishable work, researchers may increasingly rely on these systems, reducing the need for direct collaboration and communication with colleagues. This shift could lead to a decline in interpersonal interaction, weakening traditional research networks built on teamwork and shared discourse (Si et al., 2024; Hu et al., 2024b). Over time, the diminishing frequency of collaborative exchanges may foster a sense of professional isolation among human researchers, increasing the risk of loneliness, disengagement, and reduced psychological well-being.

- **Exacerbated Technological Barriers**: Without equitable access to advanced AIGS systems, a technological divide could emerge, disadvantaging researchers unfamiliar with or lacking access to these systems, thus exacerbating inequalities within the community.

**Impact on Environment.** AIGS systems can conduct large-scale experiments in parallel, but their dependence on iterative processes carries the risk of inefficient feedback loops, potentially leading to issues such as infinite loops. This inefficiency, caused by limited reasoning capabilities, the misuse of erroneous information, or ambiguity in task definition (Yang et al., 2024d), could drive up energy consumption. Moreover, poorly regulated experiments, especially without adequate simulation environments, can lead to unintended environmental harm. For example, untested chemical processes in materials science may yield hazardous by-products, while unchecked experiments in nuclear research could increase the risk of radiation leaks (Tang et al., 2024).

**Impact on Social Security.** AIGS systems, particularly when compromised by jailbreak attacks, could generate responses that conflict with human values, such as providing instructions for creating explosives. This raises concerns about their misuse for harmful purposes, such as designing more advanced adversarial attack strategies (Tang et al., 2024; Si et al., 2024; Lu et al., 2024; Kumar et al., 2024; Hu et al., 2024b). Even with benign intentions, unsupervised scientific research may introduce unforeseen societal risks. For instance, monopolizing breakthroughs in autonomous AI could lead to severe unemployment, market monopolies, and social unrest (Tang et al., 2024).

### B. Strategies for Responsible and Ethical Development of Automated Research Systems

**Strengthening the Security of Foundation Models.** The most fundamental step in mitigating security risks associated with AIGS systems is enhancing the security of their foundation models. Incorporating instructions for handling unsafe research into the alignment training corpus, alongside conducting rigorous safety audits prior to model deployment, are both crucial strategies to ensure the systems be robust and secure (Tang et al., 2024).

**Aligning Scientific Agents with Human Intentions, Environment and Self-Contraints.** Scientific agents in AIGS systems should align with human intentions, environmental dynamics, as well as self-constraints (Yang et al., 2024d).

- **Human Intentions**: Agents must accurately interpret user intent, going beyond literal language to capture the deeper purpose of scientific inquiries.
- **Environment**: Agents must adapt to the operating environments by applying domain-specific knowledge accurately and utilizing specialized tools effectively.

- **Self-Constraints**: Agents should assess the feasibility of the task, manage resources wisely, and minimize waste to ensure sustainable operation. This includes setting boundaries to prevent redundant work or harmful behavior for better system efficiency.

**Providing Comprehensive Training for Human Users.** Comprehensive and rigorous training is essential for users to fully leverage AIGS systems and prevent unintended consequences (Aidan, 2024). Proper training minimizes the risk of misuse that could lead to environmental harm, resource waste, or unethical research results (Tang et al., 2024).

**Building a Collaborative Framework Between Automated Research Systems and Human Researchers.** To prevent AIGS systems from exerting excessive influence on the academic community, collaboration between AIGS systems and human researchers will play a crucial role (Si et al., 2024; Hu et al., 2024b). It is essential to explore the new roles and responsibilities that human scientists may need to assume in this evolving research landscape shaped by the presence of AIGS systems. A well-structured partnership can leverage the complementary strengths of both, enabling outcomes that neither could achieve independently. Moreover, such collaboration fosters interaction among human researchers, encouraging deeper communication and mitigating the sense of isolation that may arise from increased reliance on automated tools.

**Establishing Comprehensive Legal and Accountability Frameworks.** A robust legal and accountability framework is crucial to govern the use of AIGS systems. This framework should:

- **Define Clear Scientific Research Boundaries**: Specify the permissible scope and limitations of the systems, and regulating agents with DSL might be helpful.
- **Clarify Responsibility and Credit Allocation**: Establish guidelines for assigning credit and responsibility for research results generated with the assistance of AIGS systems (Si et al., 2024; Hu et al., 2024b).
- **Implement Penalties for Misuse**: Outline liability measures and penalties to address harmful behavior or unethical practices involving these systems.

**Using AIGS Systems to Address Their Own Challenges** AIGS systems can also play a proactive role in addressing the challenges and even ethical issues they themselves introduce. For example, systems AIGS could be used to monitor and evaluate the results of other automated systems, identifying potential ethical issues, biases, or environmental risks before they escalate. Moreover, AIGS systems can facilitate the development of guidelines, by automating the analysis of research trends and regulatory needs, thus helping shape future policies for responsible AI use. When strategically used, AIGS systems become not only tools for discovery but also mechanisms for self-regulation, creating a virtuous cycle of innovation and governance.

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

# A. Limitations and Actionable Insights

Envisioning the future of AI-Generated Science systems powered by foundation models in the real world, in this section, we enumerate a few limitations for the current BABY-AIGS system and provide insights into the next steps of research for AIGS.

**Balance idea diversity and system executability.** As discussed in Appendix B.1, the design of the DSL enhances the system executability but may constrain the idea diversity. Achieving a balance between idea diversity and system executability requires further empirical analysis. One potential avenue is enabling agents to develop their own DSLs, which could enhance the executability of generated ideas without diminishing their diverse potential.

**Establish systematic mechanisms for evaluation and feedback.** The quality of AIGS system depends heavily on rigorous evaluation of prior proposals, methods, and results. Current approaches often adopt a peer review format, leveraging LLMs to generate feedback on results and guide future optimization (Lu et al., 2024; Yu et al., 2024; Jin et al., 2024). However, it remains unclear whether this method is the most effective for large-scale research settings. Future work should explore systematic mechanisms to analyze outcomes across iterations, maximizing experience transfer and continuous improvement.

**Strengthen the falsification procedure.** Our research underscores the importance of falsification to improve the scientific rigor of the research findings. Although we have prototyped the falsification process in our BABY-AIGS system, more efforts are required to strengthen the modules related to knowledge falsification, including the use of patterns and relationships derived from historical experiments to guide refined research proposals. In addition, it is also vital for AIGS systems to investigate whether the new scientific knowledge delivered could generalize across diverse research domains autonomously.

**Expand channels for scientific knowledge dissemination.** Facilitating the exchange of AI-Generated Science is critical, both between humans and AI and between AI systems. While Lu et al. (2024) focus on disseminating knowledge through research papers, alternative formats such as posters, podcasts, and videos are gaining traction with the rise of multimodal agents. Future research should also explore more efficient communication channels between AI systems, beyond structured text or natural language (Pham et al., 2024; Chen et al., 2024c).

**Exploring communication dynamics among autonomous AI researchers.** As discussed in Section 2, the advancement of AI-accelerated scientific discovery spans four paradigms, culminating in the emergence of an autonomous AI research community (Paradigm IV). Within this community, individual agentic researchers engage in interactions (Yang et al., 2024c) that parallel collaborative dynamics found in human scientific networks. Analyzing these communication dynamics is essential to understand how fully autonomous AI agents might effectively collaborate, exchange knowledge, and drive collective progress. In particular, a deeper exploration of these interactions in a multi-agent system will help establish communication frameworks that support optimal collaboration (Liu et al., 2024b), fostering a robust and productive AI-accelerated research community.

**Promote interdisciplinary knowledge integration and experimentation.** In this work, we primarily focused on the application of AIGS systems within the domain of machine learning, where experiments could be executed in digital worlds. However, future developments should extend these systems to address challenges in other scientific fields, such as biology, which has been preliminarily explored in a concurrent work (Swanson et al., 2024), chemistry, and physics, where cross-disciplinary knowledge integration is often crucial. One major challenge lies in how AI agents can synthesize and align domain-specific knowledge from multiple fields, which often have distinct terminologies, methodologies, and epistemological assumptions. Another critical challenge is the experiment environment, which could be hardly automated and might be highly resource-consuming.

# B. Design Details of the BABY-AIGS System

The proposed BABY-AIGS is illustrated in Figure 4, with each component detailed in the following sections.

## B.1. Domain-Specific Language (DSL)

A domain-specific language (Mernik et al., 2005) is created specifically for a particular application domain, providing greater expressiveness and ease of use within that domain compared to general-purpose languages, traditionally for programming languages. However, we observed that the situation is the same for agents in the AIGS systems. When conducting scientific research, agents have access to a wide and diverse action space, making it challenging to perform error-free long-sequence actions for every stage of the research process, particularly when translating the methodology into executable actions for

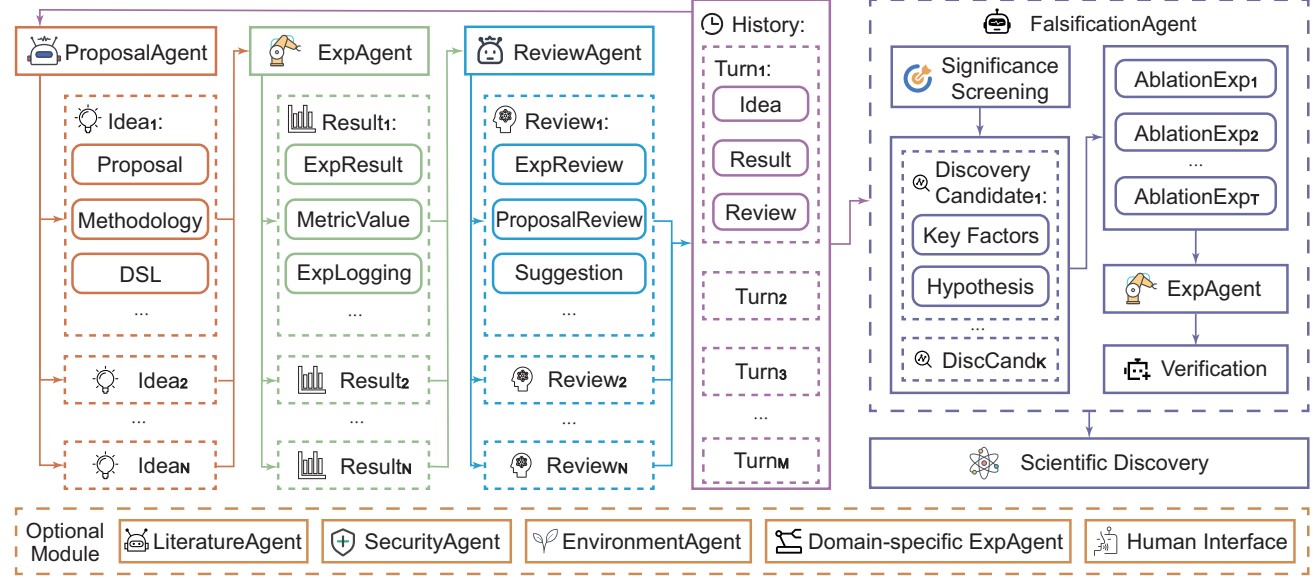

*Figure 4.* Overview of our BABY-AIGS system design. The left part denotes **Pre-Falsification** phase, where PROPOSALAGENT iteratively refine the proposed idea and methodology based on empirical and verbose feedback from EXPAGENT, REVIEWAGENT, etc. The iterative process summons multi-turn logs as the history context, based on which FALSIFICATIONAGENT could produce scientific discovery in the **Falsification** phase, as shown in the right part. Other modules are optional for the automated full-process research.

experimentation. For instance, in machine learning research, an agent may edit multiple code files and manipulate large amount of data, as part of the methodology execution. However, limited by the current capacity of foundation models, it remains a severe challenge for agents to carry out the proposed experiment with both full-process autonomy and satisfiable success rates (Jimenez et al., 2024; Chan et al., 2024; Lu et al., 2024) without dedicated interface design (Yang et al., 2024a; Wang et al., 2024) or tool use (Paranjape et al., 2023; Qin et al., 2024).

In BABY-AIGS, we extend the original definition of DSL in programming to semi-structure objects with pre-defined grammars, making it a bridge that fills the gap between the proposed methodology and experimentation. The DSL restricts the action space of the agents while maintaining the freedom for agents to conduct proposed methods at the same time, through dedicated design with human effort. To utilize the capabilities of current LLMs in natural language and function-level coding, we design the semi-structured grammar to be flexible between verbal instructions and structured statements. As shown in Figure 5, the DSL has both a higher degree of formalization and executability than natural language; compared to the coding language adopted in previous work (Lu et al., 2024), though DSL has a lower degree of formalization, with human effort, it exhibits higher executability and thus ensures successful execution of experiments, according to empirical analysis (Appendix C.4). However, when the grammar is poorly designed, the DSL is likely to restrain the creativity of the system, because some ideas might not be able to be implemented, which is a limitation of BABY-AIGS for future work.

We present the pre-defined grammar of DSL used in a few selected research topics in Figure 6. Under a specific paradigm related to the research topic, the grammar contains a series of parameters in either structured statement, e.g., code, integers, etc., or natural language, collectively depicting the methodology under the paradigm. PROPOSALAGENT would select a research paradigm when there are multiple, and fill out each parameter as required in the grammar. EXPAGENT is equipped with a pre-defined interpreter to translate the DSL into executable code lines, or inputs to specific LLMs or other models. For instance, one parameter of the DSL for data engineering is a few lines of data rating principles represented in natural language, and the model architecture parameters for language modeling still remains in codes, indicating the flexibility of DSL design. Please refer to Appendix C.1 for detailed formulation of the research topics and topic-specific DSL designs.

## B.2. PROPOSALAGENT

As the first step towards the scientific research, idea formation and methodology design usually lay the foundation for valuable insights or impactful discoveries from falsification process based on empirical results, i.e., ***creativity*** in the AIGS

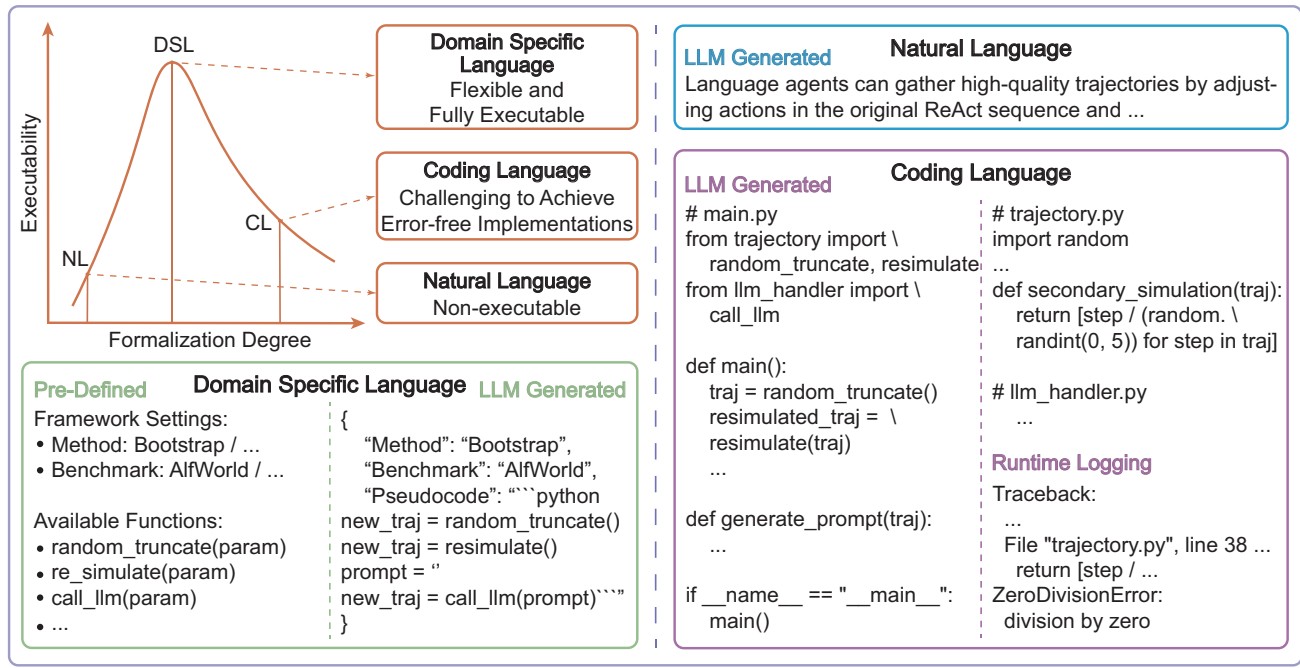

*Figure 5.* The relationship between formalization degree and system executability when expressing ideas through Natural Language (NL), Coding Language (CL), and Domain-Specific Language (DSL), illustrated with examples. NL expresses ideas in the simplest and most flexible form but is non-executable; CL offers greater precision but is challenging to achieve error-free implementation; DSL achieves a better tradeoff between flexibility and executability.

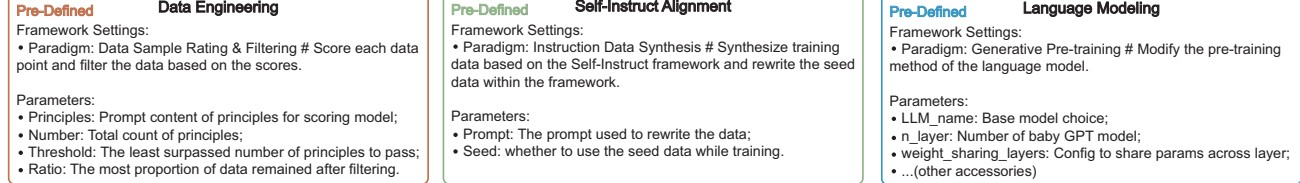

*Figure 6.* The DSL design in BABY-AIGS for open-world research topics in Appendix C.1. The full demonstration is in Appendix C.2.

system. We refer to the corresponding module in BABY-AIGS as PROPOSALAGENT, drawing inspiration from human practice of proposing an idea and formulating the methodology before starting the experiments.

PROPOSALAGENT is an important part of the pre-falsification phase. It takes the detailed description of research topic, the history log, including records of previous proposals and experiments, and the review from REVIEWAGENT as the overall input, except for the first iteration, in which only the description of the research topic is the input to PROPOSALAGENT.

Thus, the formulation of PROPOSALAGENT could be expressed as:

$$Proposal^{(i)} = \left\{ Idea\ \&\ Method.^{(i)}, Exp.\ Settings^{(i)}, Hypo.\ \&\ Related\ Feat.^{(i)}, Rebuttal^{(i)} \right\},$$
$$= \text{PROPOSALAGENT}\left( Research\ Topic\ |\ History^{(i)} \right), 1 \le i \le M, \tag{3}$$

where

$$History^{(i)} = \begin{cases} \emptyset, & \text{if } i = 1 \\ \left\{ Proposal^{(j)}, Exp.\ Results^{(j)}, Review^{(j)} \right\}_{j=1}^{i-1}, & \text{if } 1 < i \le M \end{cases}, \tag{4}$$

$i$ indicates the number of iteration, $M$ denotes the maximum iteration, PROPOSALAGENT($\cdot \mid \cdot$) indicates the agentic

*Table 3.* Examples of multi-level metrics for REVIEWAGENT to empirically review the experimental results and the proposal from PROPOSALAGENT in the data engineering research.

| Metric | Level | Description | Execution |
|---|---|---|---|
| Length
Keyword Overlap
Sentiment | Corpus | The length and word count of responses
The keyword overlap between instructions and responses
The contained sentiment in model-generated responses | Pre-defined statistic function

NLTK (Bird & Loper, 2004) |
| Worst Data Points
Best Data Points | Sample | The worst rating samples compared with baselines
The best rating samples compared with baselines | Ranking & reciting function |
| ...... | Corpus /
Sample | Other useful metrics generated by REVIEWAGENT or pre-defined by researchers | Free-form code segment |

workflow, and *Experimental Results* and *Review* are from EXPAGENT and REVIEWAGENT elaborated in Appendix B.3. The DSL format of the proposed methodology is illustrated in Appendix C.2. Building upon the aforementioned components, PROPOSALAGENT puts forward a comprehensive yet highly executable proposal, which is then submitted to EXPAGENT for execution. Upon receiving the review form REVIEWAGENT, PROPOSALAGENT can initiate the next iteration, either exploring a brand new direction or optimizing current experimental results. Examples of PROPOSALAGENT for different research topics can be found in Appendix D.5.

## B.3. REVIEWAGENT

Drawing inspiration from human practice, we recognize that significant insights and breakthroughs often emerge from in-depth analysis of experiments and reflection on methodology based on empirical results. To facilitate this process, we design REVIEWAGENT to analyze the experimental results and provide feedback to PROPOSALAGENT, iteratively improving the overall proposal.

In order to conduct a comprehensive and constructive review, REVIEWAGENT performs analysis at different levels of granularity. For fine-grained analysis, REVIEWAGENT examines comprehensive experimental logs, analyzing intermediate results from multi-level metrics which could be pre-defined by human researchers, e.g. performance indicators of the benchmark, or self-generated in code segment (examples for data engineering shown in Table 3). The *Review of the Experimental Results* identifies hidden patterns in the empirical details, resulting in fruitful low-level feedback mainly on experiment design and adjustment on the expectation of PROPOSALAGENT for the experimental results. For coarse-grained analysis, it evaluates the general validity and reasonableness of the methodology and hypothesis, providing *Review of the whole Proposal*. This review content serves as high-level advice on the idea and methodology, with the aim of provoking PROPOSALAGENT toward higher creativity.

Formally, the outcome of REVIEWAGENT could be expressed as:

$$
\begin{aligned}
Review^{(i)} &= \left\{ Review\ of\ the\ Exp.\ Results^{(i)}, Review\ of\ the\ Proposal^{(i)} \right\}, \\
&= \text{REVIEWAGENT} \left( Research\ Topic \mid Proposal^{(i)}, Exp.\ Results^{(i)}, History^{(i)} \right), 1 \le i \le M,
\end{aligned}
\tag{5}
$$

where REVIEWAGENT$(\cdot \mid \cdot, \cdot, \cdot)$ indicates the agentic workflow, and *Experimental Results* contain the benchmark results and other metric values extracted from experiments. Examples of REVIEWAGENT for different research topics can be found in Appendix D.5.

In addition, human scientists derive valuable insights not only from a literature review and reasoning, but also through empirical analysis and detailed inspection of the experimental phenomenon, especially for subjects relying largely on empirical studies. Compared to previous work (Lu et al., 2024; Su et al., 2024) that improve ideation creativity primarily based on literature, our system advances this approach by introducing multi-granular review of experimental results and processes. We argue **the groundtruth of scientific laws root and get reflected in experimental outcomes, which could serve as process supervision** in our iterative refinement of the proposal in the pre-falsification phase, and might contribute to the overall creativity of BABY-AIGS. Please refer to Appendix C.4 for empirical analysis.

**B.4. Multi-Sampling Strategy**

In this section, we formalize the multi-sampling strategy employed in the pre-falsification phase of BABY-AIGS system. This strategy is designed for better efficiency and quality of iterative exploration by parallel executing PROPOSALAGENT, EXPAGENT, REVIEWAGENT, etc. for multiple threads, combined with reranking to retain the most promising threads for further exploration.

As shown in Figure 4, the multi-sampling strategy operates orthogonal to the iterative refinement of the proposal, where the pre-falsification process of each iteration $i$ involves parallel sampling across $N$ threads, and each sampled thread represents a full pre-falsification process, including ideation, experimentation, reviewing, etc. Formally, let $\mathcal{S}^{(i)} = \{s_1^{(i)}, s_2^{(i)}, \ldots, s_N^{(i)}\}, i = 1, ..., M$ represent the set of threads sampled in iteration $i$. Each sample $s_j^{(i)}, j = 1, ..., N$ undergoes experiments and reranking based on pre-defined criteria, and only a subset with top-ranked samples $\mathcal{S}_{\text{top}}^{(i)} \subset \mathcal{S}^{(i)}$ of size $N_s$ is retained for the next iteration. The process can be summarized as follows:

1. **Sampling Step**: In each iteration $i$, the system generates $N$ samples $\{s_1^{(i)}, s_2^{(i)}, \ldots, s_N^{(i)}\}$ in parallel. If the former samples $\mathcal{S}_{\text{top}}^{(i-1)}$ are available, i.e., it is not the first iteration, each $s_j^{(i)}, j = 1, ..., N$ is generated by taking into account the historical log from the $\left(j\lfloor \frac{N}{N_s} \rfloor + 1\right)$-th sample of the previous $\mathcal{S}_{\text{top}}^{(i-1)}$ threads, i.e. $s_{j\lfloor \frac{N}{N_s} \rfloor + 1}^{(i-1)}$.

2. **Reranking**: All samples are reranked on the basis of the benchmarking result during experimentation. For simplicity, we adopt the average performance score of all benchmarks.

3. **Selection for Next Iteration**: After step 2, the samples are reranked and the top $N_s$ samples are selected to form the set $\mathcal{S}_{\text{top}}^{(i)}$ for the next iteration.

Within BABY-AIGS, the multi-sampling strategy with reranking is applied primarily in the **Pre-Falsification** phase, facilitating an extensive yet efficient exploration of ideas, methods, and experimental configurations. By iteratively narrowing down to the top candidates, this strategy effectively focuses resources on promising pathways. In Appendix C.5, we empirically demonstrate the multi-sampling strategy, coupled with reranking, is essential for guiding the iterative process in BABY-AIGS towards scientifically significant discoveries in an effective and potentially scalable manner.

**B.5. FALSIFICATIONAGENT**

In the research process, there is usually a gap between the experimental results indicating improvement in performance and the final conclusions of the scientific findings, and human researchers usually perform ablation studies to verify the authenticity of scientific discoveries. We term progress like this *falsification*, which is a critical step towards full-process automated scientific discoveries.

Recognizing the importance of *falsification*, we introduce FALSIFICATIONAGENT, a novel component not present in previous work (Lu et al., 2024; Su et al., 2024). FALSIFICATIONAGENT has access to all history records, including proposals from PROPOSALAGENT, experiment results from EXPAGENT, and reviews from REVIEWAGENT. We hypothesize that important scientific discoveries are more likely to emerge from significant experimental phenomena, i.e. changes in results, thus, FALSIFICATIONAGENT in BABY-AIGS first performs a "Significance Screening" to identify adjacent turns from pre-falsification phase with greatest performance discrepancies, as shown in Figure 7. Following this, FALSIFICATIONAGENT generates scientific discovery candidates from these selected turns. Then FALSIFICATIONAGENT generates the plans and the ablated methods for ablation experiments. We require that at most $T$ plans are made for each discovery candidate, indicating that at most $T$ ablation experiments will be conducted, and each ablation experiment focuses on the verification of a single factor that may influence the experimental result. Specifically, FALSIFICATIONAGENT must select an iteration from pre-falsification as the baseline for the ablation study, and FALSIFICATIONAGENT follows the "Experiment Settings" of the baseline, and modify the methodology according to the ablated factor.

Attempting to reach a robust and reliable conclusion of the ablation study, both baseline and ablation experiments are repeated multiple times. FALSIFICATIONAGENT is given the complete record of these experiments to decide the validity of the associated scientific principle. If a particular discovery withstands this process and consistently produces results similar to those in the main experiment, it is regarded as a verified and valuable *Scientific Discovery*. And it is falsified otherwise.

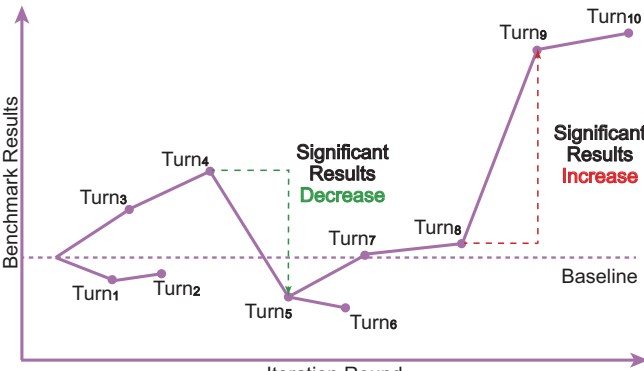

*Figure 7.* Illustration of "Significance Screening" on the history records of pre-falsification phase. The node of each turn represents the experimental results from the modified method. The "Significance Screening" process identifies results with significant performance increase or decrease for probably important scientific discoveries.

Formally, the outcome of FALSIFICATIONAGENT, which is also the output of BABY-AIGS, is:

$$Scientific\ Discovery = \text{FALSIFICATIONAGENT}\left(Research\ Topic \mid History\right), \tag{6}$$

where

$$History = \left\{Proposal^{(i)}, Exp.\ Results^{(i)}, Review^{(i)}\right\}_{i=1}^{M}, \tag{7}$$

and FALSIFICATIONAGENT$(\cdot \mid \cdot)$ indicates the agentic workflow. Examples of FALSIFICATIONAGENT for different research topics can be found in Appendix D.5.

To our knowledge, FALSIFICATIONAGENT is the first agent within AI-accelerated scientific discovery systems capable of autonomously completing the falsification process, by independently proposing scientific discovery candidates, designing and executing ablation experiments, and performing verification. For a detailed qualitative analysis, see Section **??** and Appendix C.4.

## C. Automated Full-Process Research Experiment

We conduct experiments on four primary research topics in machine learning to evaluate BABY-AIGS in autonomous full-process research. Formally, let $\mathcal{D}_k = \{(x_i, y_i)\}_{i=1}^{N}$ denote the $k$-th benchmark of a given ML problem, where $x_i$ represents input features and $y_i$ represents the corresponding labels. The goal is building a system $f : \mathcal{X} \to \mathcal{Y}$ that maximizes metric functions $\mathcal{L}_k(f(x), y)$ over all benchmark $\mathcal{D}_k$. We split benchmarks into validation and test ones, and only the former is available in the pre-falsification phase, avoiding wrong scientific discoveries from over-fit results.

### C.1. Selected Research Tasks

#### C.1.1. OPEN-WORLD RESEARCH TOPICS

**Data Engineering**  Data engineering is a critical research topic that focuses on the identification, extraction, and processing of relevant data features that significantly influence model performance. We formulate the research goal as follows: Given a data set $\mathcal{H}$ that contains instruction-response pairs, the goal is to identify the key distinguishing characteristics of $\mathcal{H}$, which in turn enables the system to filter and extract high-quality data subsets $\mathcal{H}' \subset \mathcal{H}$ for the development of LLMs. This process is crucial to improving the quality and relevance of data for a wide range of areas, ensuring downstream tasks, such as in-context learning (Brown et al., 2020) and Supervised Fine-Tuning (SFT) for LLM alignment (Ouyang et al., 2022), are more effective. Specifically, we leverage Alpaca-GPT4 dataset (Peng et al., 2023) as the dataset $\mathcal{H}$. We follow previous work (Liu et al., 2024a; Chen et al., 2024b; Li et al., 2024d; Zhao et al., 2024) in this field and let the AIGS systems write principles for LLMs to rate data samples and extract the top rated ones as the refined dataset. Thus, for BABY-AIGS, we input the description of the topic and design the main DSL as a list of required principles for the evaluation of the data sample and a threshold indicating the least number of principles that a data sample in the refined dataset has to pass.

**Self-Instruct Alignment** The self-instruct alignment (Wang et al., 2023c) is a well adopted data synthesis paradigm for LLM alignment. The objective of this research topic is to synthesize a set of SFT data with high quality and diversity for LLM alignment (Ouyang et al., 2022) by rewriting a seed set of data, thereby enhancing the performance of the fine-tuned model on this dataset. In the research process, an AIGS system is required to construct an optimal set of instructions from a seed instruction dataset, which are used to generate an instruction-response dataset from LLMs. This dataset is then leveraged to refine the alignment of an LLM via SFT. In the experiment, we rewrite the original seed instruction set, and use the same LLM in instruction synthesis and response generation for SFT data. Specifically, for BABY-AIGS, the DSL is designed as an option whether to use the seed instruction set, and a list of requirements for the given LLM to generate instructions.

**Language Modeling** Language modeling is a core research topic in natural language processing that aims to improve the ability of a model to understand and generate human language. Currently, the mainstream approach is generative pre-training (Radford et al., 2018), and the objective is to maximize the perplexity of the next token prediction, i.e. minimize the model perplexity. The AIGS system seeks to explore different architectural and training schedule modifications to enhance quality of language model pre-trained on large corpora. We designed DSL of the BABY-AIGS system as a set of constrained configurations of model architecture and training hyper-parameters.

C.1.2. CLOSED-WORLD ENVIRONMENTS

**DiscoveryWorld++** The aforementioned three research topics are all open-world tasks that aim at solving critical problems in machine learning research. Also, we introduce DiscoveryWorld++, a closed-world research environment modified from DiscoveryWorld (Jansen et al., 2024). To be specific, we select Plant Nutrients and Chemistry, two research topics that inherently involve the process of falsification in order to reach the true scientific discovery. In Plant Nutrients task, the agent is required to figure out the certain nutrients with certain level of amount in the soil that can promote the growth of mushrooms. In DiscoveryWorld, only one nutrient is positive for the growth, while in our DiscoveryWorld++, two kinds of nutrients can help with the growth and the agent needs to answer both two nutrients. In Chemistry task, the agent is required to remove the rust attached to the key by mixing four chemicals and figure out the mixture with least amount of each chemical that can effectively remove rust. In DiscoveryWorld, the agent only needs to remove rust, which means that it can mix all chemicals together in order to finish the task. In contrary, in DiscoverWorld++, it must answer the least amount of each chemical in the mixture, which consequently requires falsification. Moreover, In DiscoveryWorld, once all the sub-tasks are finished, the task is marked as success even if the agent does not mean to quit the loop and just reach success out of expectation. In DiscoveryWorld++, the agent can explicitly quit the task with submitting an answer to the core question of the task, which means that it has finished all the explorations and experiments and has been sure about its answer. In this way, we can conduct a more comprehensive evaluation on the performance of the agent as well as the process of falsification.

Each of these research topics requires unique methodological innovations of an AIGS system to foster high *creativity*, *executability*, and *falsification* capabilities. We demonstrate the pre-defined grammars of BABY-AIGS in Figure 6. Please refer to Appendix C.2 for detailed settings.

**C.2. Implementation Details of the BABY-AIGS system**

In this section, we elaborate the implementation details of the BABY-AIGS system. All artifacts are used as intended with their license strictly followed in our work.

C.2.1. RESEARCH-AGNOSTIC IMPLEMENTATION

**System Pipeline** We posit that all agents mentioned in Section 5.1 contribute to a full-process AIGS system, but based on preliminary experiments, we simplify the design of EXPAGENT and LITERATUREAGENT to a large extent in our implementation. For EXPAGENT, given the design of DSL with human effort, proposed methodology generated by PROPOSALAGENT can be executed reliably in experiments, which is also shown in Table 10. This reduces the need of iteratively refining proposals between PROPOSALAGENT and EXPAGENT. For LITERATUREAGENT, preliminary results show literature integration did not significantly impact the outcomes in both phases of BABY-AIGS. We conclude the reason as that agents failed to understand the in-depth literature information and the retrieval of literature did not match the need of each agent perfectly. Therefore, in our implementation, we minimize the design of these two agents: EXPAGENT functions through fixed code, and LITERATUREAGENT was not put into pratical use. Other optional agents are designed to function in

broader research fields, and we chose to omit them in experiments based on the selected research topics for experiments (Appendix C.1).

**Hyper-Parameters**  Experiments in ICL (In-Context Learning) of the data engineering research and in language modeling research are conducted on 8 NVIDIA GeForce RTX 3090 24 GB GPUs. Experiments in SFT (Supervised Fine-tuning) of the data engineering research and in Self-Instruct alignment research are conducted on 8 A100 80GB GPUs. All researches utilize the gpt-4o-2024-05-13 model as the underlying model for our agents. When agents invoke GPT-4o, we use the openai module[1] with a temperature setting of 0.7, while all other parameters are setting as default values. During the synthesis of proposals, PROPOSALAGENT generates three sets of proposals with a temperature of 0.7. After generation, the Jaccard similarity (Jaccard, 1901) of bigram sets is calculated between the methodology of each proposal and the methodology produced in the previous iteration. The proposal with the lowest similarity in methodology is selected as the final output to increase its diversity. For REVIEWAGENT and FALSIFICATIONAGENT, they invoke the GPT-4o only once each time when generating responses.

### C.2.2. OPEN-WORLD RESEARCH-SPECIFIC IMPLEMENTATION

**Data Engineering**  In this research experiment, our system is tasked with exploring different approaches to improve the quality of Alpaca-GPT4 dataset (Peng et al., 2023). The DSL configuration and instance are shown in Figure 6 and Figure 8. The Llama-3-8B-Instruct[2] model is employed to rate all data samples with the principles in DSL. We deploy Llama-3-8B-Instruct using vLLM[3], configuring the temperature to 0.05, while keeping all other parameters at the default settings. We use Llama-3-8B[4] for ICL- and SFT-alignment, and the model and the fine-tuned checkpoints are deployed using vLLM with a maximum token limit of 1024, while other parameters follow the default configurations provided by FastChat[5]. In falsification process, the BABY-AIGS system identifies the factors that contribute to quality improvements and conclude whether there are ways to stably improve the quality of the extracted dataset, thus delivering valuable scientific discoveries. For significance screening in FALSIFICATIONAGENT, iterations are identified as having significant improvements if the difference of adjacent benchmarking results exceeds 1.5 for the ICL-aligned Llama-3-8B on the Vicuna-Bench (the validation benchmark) or 0.5 on the MT-Bench (the test benchmark). From these iterations, candidates for scientific discovery are extracted. For hyper-parameters, we set the total iteration number $M = 5$ and set the multi-sample threads number $N = 32$.

**Self-Instruct Alignment**  In this research experiment, our system is tasked with exploring different approaches to improve the quality of synthesized SFT data from a seed dataset in Self-Instruct[6] (Wang et al., 2023c). We use GPT-4o to rewrite the seed data for better quality with the temperature parameter set to 0.05. The DSL configuration and instance are shown in Figure 6 and Figure 9. We use the Llama-3-8B[7] model to generate instructions and responses, with it also serving as the base model for SFT alignment. We use LoRA (Hu et al., 2022) method from LLaMA-Factory[8] to fine-tune the model with default training hyper-parameters[9]. The other experiment setting is the same as data engineering research. For hyper-parameters, we set the total iteration number $M = 15$ and set the multi-sample threads number $N = 1$ due to limited computing resource.

**Language Modeling**  In this research experiment, our system is tasked to pre-train a mini-sized language model on several small corpora, aiming to improve performance by minimizing loss on the selected datasets. The experiment mainly follows the same setup as the language modeling task in AI Scientist (Lu et al., 2024), based on the nanoGPT project [10]. The DSL configuration and instance are shown in Figure 6 and Figure 10, where we guide the models in adjusting parameters related to model architecture and training process. For the experiments, we use the sampling scripts provided in the template code without modifications. For hyper-parameters, we set the total iteration number $M = 10$ and set the multi-sample threads number $N = 1$ due to limited computing resources for parallel model training.

---

[1]https://github.com/openai/openai-python
[2]https://huggingface.co/meta-llama/Meta-Llama-3-8B-Instruct
[3]https://github.com/vllm-project/vllm.
[4]https://huggingface.co/meta-llama/Meta-Llama-3-8B.
[5]https://github.com/lm-sys/FastChat.
[6]https://github.com/yizhongw/self-instruct.
[7]https://huggingface.co/meta-llama/Meta-Llama-3-8B.
[8]https://github.com/hiyouga/LLaMA-Factory.
[9]https://github.com/hiyouga/LLaMA-Factory/blob/main/examples/train_lora/llama3_lora_sft.yaml.
[10]https://github.com/karpathy/nanoGPT.

*Table 4.* Statistic results of human evaluation on the falsification process in our data engineering research experiments.

| Metric | AVG | STD | P-Value | MIN | MAX |
|---|---|---|---|---|---|
| **Importance Score** (0 ∼ 2) | | | | | |
| BABY-AIGS (Ours) | 1.80 | 0.41 | 0.02 | 0.00 | **2.00** |
| Top Conference | **2.00** | 0.00 | — | **2.00** | **2.00** |
| **Consistency Score** (0 ∼ 2) | | | | | |
| BABY-AIGS (Ours) | 1.00 | 0.86 | 0.00 | 0.00 | **2.00** |
| Top Conference | **2.00** | 0.00 | — | **2.00** | **2.00** |
| **Correctness Score** (0 ∼ 2) | | | | | |
| BABY-AIGS (Ours) | 0.95 | 0.94 | 0.00 | 0.00 | **2.00** |
| Top Conference | **2.00** | 0.00 | — | **2.00** | **2.00** |
| **Overall Score** (0 ∼ 2) | | | | | |
| BABY-AIGS (Ours) | 1.25 | 0.47 | 0.00 | 0.67 | **2.00** |
| Top Conference | **2.00** | 0.00 | — | **2.00** | **2.00** |

*Table 5.* Statistic results of human evaluation on the falsification process in our self-instruct alignment research experiments.

| Metric | AVG | STD | P-Value | MIN | MAX |
|---|---|---|---|---|---|
| **Importance Score** (0 ∼ 2) | | | | | |
| BABY-AIGS (Ours) | 1.60 | 0.50 | 0.00 | 1.00 | **2.00** |
| Top Conference | **2.00** | 0.00 | — | **2.00** | **2.00** |
| **Consistency Score** (0 ∼ 2) | | | | | |
| BABY-AIGS (Ours) | 1.15 | 0.49 | 0.00 | 0.00 | **2.00** |
| Top Conference | **2.00** | 0.00 | — | **2.00** | **2.00** |
| **Correctness Score** (0 ∼ 2) | | | | | |
| BABY-AIGS (Ours) | 0.85 | 0.59 | 0.00 | 0.00 | **2.00** |
| Top Conference | **2.00** | 0.00 | — | **2.00** | **2.00** |
| **Overall Score** (0 ∼ 2) | | | | | |
| BABY-AIGS (Ours) | 1.20 | 0.35 | 0.00 | 0.33 | **2.00** |
| Top Conference | **2.00** | 0.00 | — | **2.00** | **2.00** |

*Table 6.* Statistic results of human evaluation on the falsification process in our language modeling research experiments.

| Metric | AVG | STD | P-Value | MIN | MAX |
|---|---|---|---|---|---|
| **Importance Score** (0 ∼ 2) | | | | | |
| BABY-AIGS (Ours) | 1.75 | 0.55 | 0.03 | 0.00 | **2.00** |
| Top Conference | **2.00** | 0.00 | — | **2.00** | **2.00** |
| **Consistency Score** (0 ∼ 2) | | | | | |
| BABY-AIGS (Ours) | 1.20 | 0.62 | 0.00 | 0.00 | **2.00** |
| Top Conference | **2.00** | 0.00 | — | **2.00** | **2.00** |
| **Correctness Score** (0 ∼ 2) | | | | | |
| BABY-AIGS (Ours) | 1.10 | 0.79 | 0.00 | 0.00 | **2.00** |
| Top Conference | **2.00** | 0.00 | — | **2.00** | **2.00** |
| **Overall Score** (0 ∼ 2) | | | | | |
| BABY-AIGS (Ours) | 1.35 | 0.46 | 0.00 | 0.33 | **2.00** |
| Top Conference | **2.00** | 0.00 | — | **2.00** | **2.00** |

### C.2.3. CLOSED-WORLD RESEARCH-SPECIFIC IMPLEMENTATION

**DiscoveryWorld++**   In this reasearch environment, out system is tasked with two tasks: Plant Nutrients and Chemistry. The agent needs to explore in the environment, design and conduct the experiments, and finally answer the question related to the scientific discovery based on observations in the environment and the experimental results. Considering that the actions provided in DiscoveryWorld are specially designed for agent execution and fit the needs of environment and tasks, we directly use these defined actions as DSL for our BABY-AIGS. In DiscoveryWorld++, we set the total iteration number $M = \min(300, M_s)$, where $M_s$ refers to the iteration that the agent decides to submit the answer to quit loop. Also, as each iteration step does not always lead to a experimental result which is helpful to the task progress (e.g. turn to a certain direction; move forward), and no randomness of experiment is involved in the environment, the multi-sample threads number $N$ is set as 1 for DiscoveryWorld++. For fair comparison, we follow the hyper-parameter settings including seed and temperature of DiscoveryWorld (Jansen et al., 2024) and employ GPT-4o for all the agents in our experiment.

### C.3. Evaluation Settings

**Falsification**   The human evaluation results of the three open-world topics can be found in Table 4, Table 5 and Table 6 respectively.

**Creativity**   We measure the creativity of BABY-AIGS by evaluating the performance improvement of the proposed idea and methodology against the baseline result, i.e., the result from the trivial methodology on the test benchmarks. Here are the benchmark settings for each open-world research topics:

- **Data Engineering**: For the refined dataset, we conduct 15-shot In-Context Learning (ICL) (Jiang et al., 2024) and SFT for LLM alignment to evaluate the overall quality. We evaluate the ICL-aligned LLM on the Vicuna-Bench, as an efficient validation benchmark, and ICL- and the SFT-aligned LLM on the MT-Bench (Zheng et al., 2023), which are used as test benchmarks. The baseline of turn 0 uses the original Alpaca-GPT4 dataset (Peng et al., 2023). We replicate AI Scientist with the same experiment template. Moreover, we replicate Deita (Liu et al., 2024a) as the human research of the topic from the top conference.

- **Self-Instruct Alignment**: We also assess the aligned LLM on the Vicuna-Bench, as the validation benchmark, and the MT-Bench, as the test benchmark. The baseline of turn 0 is the result of the original self-instruct method (Wang et al., 2023c).

- **Language Modeling**: We pre-train a mini-sized language model with the modified architecture based on the configured training schedule, on three different training sets (Karpathy, 2015; Hutter, 2006; Mahoney, 2011). The validation and test benchmarks are the perplexity of LM on the split validation and test sets. With reference to Lu et al. (2024), we adopt the default settings of the nanoGPT project[11] as the baseline.

Results on all test benchmarks are in Table 7, Table 8, and Table 9, for each topic, respectively.

**Executability**   We evaluate the BABY-AIGS system's stability to execute research ideas errorlessly from ideation to implementation, measured by the success rate of obtaining meaningful experimental outcomes and scientific insights, termed as Experiment Success Rate (Exp. SR) and Overall Success Rate (Overall SR), respectively. We report the overall results on all research experiments on the three topics. AI Scientist as the baseline method, are also evaluated executability on the selected tasks in their original implementation (Lu et al., 2024). Results are shown in Table 10.

### C.4. Quantitative and Qualitative Analysis

**BABY-AIGS demonstrates creativity during research idea exploration and refinement.**   Table 7, Table 8, and Table 9 show the results of the test benchmarks for *data engineering*, *self-instruct alignment*, and *language modeling* research topics, respectively, where BABY-AIGS outperforms the baseline method, demonstrating the system's creativity in ideation and corresponding method design. For data engineering, BABY-AIGS outperforms AI Scientist with a significant margin, demonstrating the effectiveness of the enriched feedback, including multi-granular metrics, verbose review on both experiment process and methodology design, etc., in exploring research idea. However, the result of SFT alignment is

---

[11] https://github.com/karpathy/nanoGPT.

*Table 7.* Benchmarking results on the test benchmarks of the data engineering research experiment (left) and a summarization of the corresponding proposed methodology from BABY-AIGS (right).

| Method | MT-Bench ↑ | |
|---|---|---|
| | 15-shot ICL | SFT |
| Baseline (Turn 0) | 4.18 | 4.53 |
| AI Scientist | 4.36 | 4.67 |
| **BABY-AIGS (Ours)** | **4.51** | 4.77 |
| Top Conference | 4.45 | **5.01** |

**Methodology Summarization (Data Engineering)**

1. Rate the response based on its contextual coherence, ensuring it logically follows the conversation.
2. Evaluate the relevance by checking if the answer stays on-topic with minimal digression.
3. Check for logical reasoning in explanations, ensuring the response is not just factual but also thoughtful.
4. Consider if the complexity and detail match the question's requirements, avoiding oversimplification.
5. Finally, evaluate the tone for politeness, clarity, and natural conversational flow.

*Table 8.* Benchmarking results on the test benchmark of the self-instruct alignment research experiment (left) and a summarization of the corresponding proposed methodology from BABY-AIGS (right).

| Method | MT-Bench ↑ |
|---|---|
| Baseline (Turn 0) | 2.28 |
| **BABY-AIGS (Ours)** | **3.025** |

**Methodology Summarization (Self-Instruct Alignment)**

Make the instruction to cover different scenarios if it lacks specificity, clearer if ambiguous, aligned with natural conversations, and to contain a diverse range of task types if it lacks variety.

*Table 9.* Benchmarking results on the test benchmarks of the language modeling research experiment (left) and a summarization of the corresponding proposed methodology from BABY-AIGS (right).

| Method | Perplexity ↓ | | |
|---|---|---|---|
| | shakespeare_char | enwik8 | text8 |
| Baseline (Turn 0) | **1.473** | 1.003 | 0.974 |
| **BABY-AIGS (Ours)** | 1.499 | **0.984** | **0.966** |

**Methodology Summarization (Language Modeling)**

Reduce the dropout rate with more attention heads to increase model expressiveness. And implement a cyclical learning rate and adjust the weight decay to regularize the model.

inferior than Deita (Liu et al., 2024a), indicating that the lack of validation benchmarking of specific downstream tasks might result in an suboptimal outcome.

**BABY-AIGS has remarkable executability in experimentation and full research process.** As shown in Table 10, our quantitative analysis highlights significant improvements in executability, with BABY-AIGS achieving nearly 100% success rates in translating the generated ideas into experimental results and the final scientific discovery. This high executability, attributed to our DSL design for errorless experimentation, prevents restarting from in-process failures and enables an efficient automated research process. Detailed API costs are elaborated in Appendix D.1.

### C.5. Discussions

**Q1: How do current LLMs perform in the falsification process?** Falsification (Popper, 1935) is essential in AIGS systems as it provides a rigorous mechanism for verification of potential scientific discoveries, a core component in the scientific method. In BABY-AIGS, FALSIFICATIONAGENT plays the corresponding role. Thus, it demands related abilities in the foundation model, such as reasonable hypothesis generation, ablation experiment design, summarization and self-correction based on input empirical results, etc. As shown in the case in Appendix B.5 and Table 2, current LLMs are far from desired in the agentic workflow of FALSIFICATIONAGENT. Additionally, the constraints may come from the ability of the LLM to understand the environment outside FALSIFICATIONAGENT. For instance, from our observation, FALSIFICATIONAGENT seldom proposes experiment plans beyond the provided experiment templates. In this case, although DSL makes sure the executability of the experimentation by omitting extra operations, the experiment process would differ from the original plan, thus creating inconsistency.

*Table 10.* Success rates on three selected tasks of AI Scientist and Baby-AIGS. Exp. SR denotes the times a system successfully conducted experiments out of all trials, and Overall SR denotes the times a system produces the final scientific discoveries. Higher numbers indicate better executability.

| Method | Experiment Success Rate (Exp. SR) | Overall Success Rate (Overall SR) |
|---|---|---|
| AI Scientist | 44.8% | 29.2% |
| **Baby-AIGS (Ours)** | **Almost 100%** | **Almost 100%** |

*Table 11.* Results on MT-Bench (15-shot ICL) of the ablation study on the multi-sampling strategy of our BABY-AIGS system in the data engineering research experiment. $N$ in "Multi-Sampling@N" indicates the number of parallel threads of multi-sampling.

| Method | Baseline | Turn 1 | Turn 2 | Turn 3 | Turn 4 | Turn 5 |
|---|---|---|---|---|---|---|
| Multi-Sampling@1 | 4.18 | 3.68 | 4.01 | 4.05 | 3.88 | 3.90 |
| Multi-Sampling@32 | 4.18 | 4.02 | 4.05 | 4.50 | 4.51 | 4.42 |

**Q2: Could REVIEWAGENT serves as the FALSIFICATIONAGENT in the BABY-AIGS system?** Previous work (Lu et al., 2024; Su et al., 2024; Weng et al., 2024) typically involve an iterative process in research ideation and methodology refinement, along with designs similar to REVIEWAGENT. This iterative exploration of research ideas and methods is indispensable. However, the review process of the changes in the methodology and the difference in the corresponding experimental results could not replace the explicit falsification process. In practice, we observed that behaviors of the exploration on the methodology of the AI Scientist and the pre-falsification phase of BABY-AIGS are varied in a wide range, from a subtle adjustment of hyper-parameters to an abrupt rewriting of the whole idea. In quite a few cases, the changes of the experimental results resulting from the refinement of methodology could not represent clear single-factor patterns or scientific discoveries without dedicated ablation experiments, except for few instances. As a high-level explanation, we argue that *an efficient and effective research process does not need to analyze the details of each possible change in methodology that has a random impact, but should analyze in detail those important changes that could possibly have a significant impact*.

**Q3: How does the BABY-AIGS system boost creativity?** BABY-AIGS enhances creativity by integrating a multi-sampling approach combined with re-ranking, allowing it to generate diverse research proposals and rank them based on validation benchmarks. We provide detailed results of an ablation study of this process in Table 11. We observed that the performance on the test benchmark is steadily increasing with multi-sampling with large numbers of threads. This strategy is related to search-based inference-cost scaling methods (Snell et al., 2024; Brown et al., 2024). The insight is to pick random high-performing samples for better overall performance. However, since the objective of AIGS is to discover science on a research topic, the reranking method here could be large-scale validation benchmarks indicating generalization performance, rather than reward-model-based (Stiennon et al., 2020) or self-verification methods for a specific query. As depicted in Appendix B.3, we argue that the groundtruth of scientific laws is rooted and reflected in benchmarking results from actual experiments, which could serve as process supervision, which could be more accurate than reward models. It explains how collapse in self-refinement-style methods (Xu et al., 2024) is avoided in this setting, which is also empirically validated through the ablation results.

**Q4: Why could DSL help with executability?** The use of a Domain-Specific Language (DSL) in BABY-AIGS facilitates executability by providing a structured and executable representation of ideas and methodologies proposed by PROPOSALAGENT. DSL enhances the system's ability to translate complex scientific workflows into actionable experiment plans. As shown in Table 10, DSL significantly improved success rates in generating scientific discoveries, regardless of correctness, underscoring its role in achieving high executability. We acknowledge that the design of DSL requires human effort and might not be able to cover all possible method implementations. However, we believe it is a promising interface between agents and experimentation in full-process research.

# D. Experiment Details

## D.1. API Costs of the Full-Process Research Experiment

In our experiments, we measured the average token counts and costs of different phases of BABY-AIGS (Section 5.1) for invoking the GPT-4o API and the results are presented in Table 12. Note that as the experimental records in past iterations are used as input in most requests, with the rise of iteration, the length of record will consequently increase, leading to the use of more tokens.

*Table 12.* Average token consumption and API costs for GPT-4o API in the full-process research experiment. The costs at pre-falsification phase is calculated for each iteration, and the costs at falsification phase is calculated for each discovery candidate.

|  | Input Tokens | Generated Tokens | Cost ($) |
| --- | --- | --- | --- |
| **Pre-Falsification** (per iter.) | 6,616.2 | 761.5 | 0.045 |
| **Falsification** (per disc. cand.) | 43,375.5 | 1,120.3 | 0.234 |

## D.2. DSL Demonstrations for Different Research Topics

We present an example of the methodology in DSL format generated during the experiment for each research topic, as shown in Figure 8, Figure 9 and Figure 10, corresponding to data engineering, self-instruct alignment, and language modeling, respectively.

---

**LLM Generated**                               Data Engineering

```
{
    "Paradigm": "Data Sample Rating & Filtering",
    "Principles": "Relevance: The data point should be relevant to the task of multi-turn conversation and instruction-following.\n
Completeness: The conversation should be complete and coherent, without abrupt endings or missing context.\nInstruction adherence:
The responses should strictly follow the given instructions and provide accurate information.\nConversational flow: The dialogue should
flow naturally, with each turn logically following the previous one.\nComplexity handling: The data point should demonstrate the ability to
handle complex and challenging questions effectively.\nEngagement: The conversation should be engaging and maintain the user's
interest throughout.\nEach data point should involve multi-turn dialogue.\nResponses should be contextually relevant to the preceding
turns.\nData points should cover a diverse range of topics to mimic common use cases.\nResponses should be coherent and
grammatically correct.\nEach response should show logical progression and consistency across turns.\nRelevance to the prompt: The
response should directly address the question or task presented in the prompt.\nCoherence: The response should be logically structured
and easy to follow.\nGrammar and syntax: The response should be free of grammatical and syntactic errors.\nCreativity and depth: The
response should demonstrate creative thinking and provide in-depth information when required.\nConsistency: The response should
maintain consistency in its argument or narrative throughout.\nLength: Ensure responses are comprehensive, aiming for lengths similar
to high-scoring entries (1000 to 3000 characters).\nWord Count: Encourage comprehensive and thorough responses, ensuring the
content is relevant and informative.\nUnique Words: Ensure responses contain a broad range of unique words while maintaining
relevance and coherence.\nStopwords Count: Ensure responses are detailed and contextually rich.\nKeyword Overlap: Ensure
responses are relevant and contextually appropriate.\nDiversity: Aim for answer diversity in the range of 0.396 to 0.690.\nAverage Word
Length: Encourage balanced word lengths between queries and answers.\nSentiment: Train models to deliver engaging, relevant, and
positive responses.\nCoherence Score: Refine the scoring method to better capture logical progression and consistency.\nInstruction
Adherence: Ensure responses have high instruction adherence.\nComplexity Score: Prioritize generating detailed and complex answers.
\nEngagement Score: Ensure responses are engaging and interactive.",
    "Number": 27,
    "Threshold": 15,
    "Ratio": 0.7
}
```

*Figure 8.* The DSL instance for data engineering research.

---

**Self-Instruct Alignment**

LLM-Generated

{
   "Paradigm": "Instruction Data Synthesis",
   "Prompt": "1. Ensure queries are between 50-150 characters and answers are between 300-1500 characters. Aim for clear and concise queries (10-26 words) and detailed yet concise answers (55-254 words).\n2. Balance specificity to provide clear and relevant information without being overly detailed (Query specificity: 1, Answer specificity: 2-4). Ensure specific terms are contextually relevant.\n 3. Maintain moderate complexity in language to ensure clarity and conciseness (Query clarity score: 2-5, Answer clarity score: 3-7). Avoid jargon unless necessary.\n4. Increase relevance by incorporating task-specific keywords and ensuring both queries and answers are contextually relevant and detailed. Ensure answers directly address the queries.\n5. Diversify the seed data to cover a broad range of tasks, topics, and scenarios, including more complex instructions. Include tasks of varying complexity and from different domains (e.g., healthcare, finance, education).\n6. Use an LLM to perform the initial evaluation and rewrite. Have human reviewers refine the rewritten instructions.\n7. Implement a structured feedback mechanism to continuously refine the principles and methodology.\n 8. Analyze high-scoring tasks and responses on VicunaBench and MT-bench to tailor the principles.",
   "Seed": true
}

*Figure 9.* The DSL instance for self-instruct alignment research.

---

**Language Modeling**

LLM Generated

{
   "Paradigm": "Generative Pre-training",
   "LLM_name": "gpt-4o",
   "n_layer": 6,
   "n_embd": 384,
   "dropout": 0.2,
   "bias": false,
   "learning_rate": 0.001,
   "max_iters": 5000,
   "weight_decay": 0.1,
   "beta1": 0.9,
   "beta2": 0.99,
   "grad_clip": 1.0,
   "decay_lr": true,
   "warmup_iters": 100
   "lr_decay_iters": 15,
   "min_lr": 0.0001
}

*Figure 10.* The DSL instance for language modeling research.

## D.3. Prompting Structure

In this section, we will briefly introduce the prompting structures of the PROPOSALAGENT, REVIEWAGENT, and FALSIFI-CATIONAGENT as shown in Figure 11, Figure 12, and Figure 13, respectively.

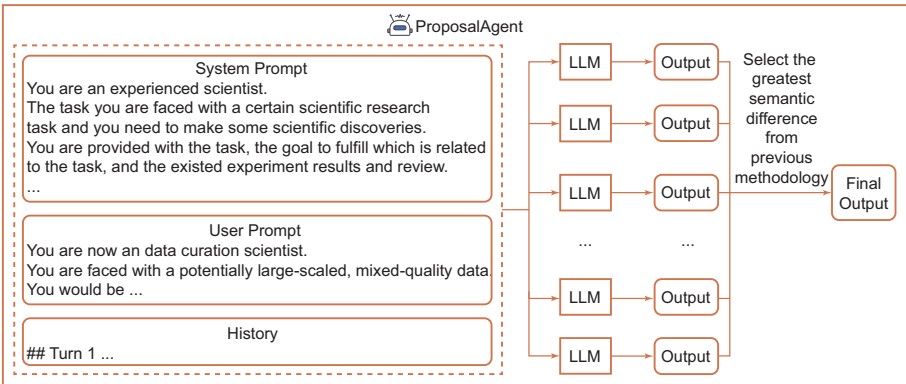

*Figure 11.* The prompting structure for the PROPOSALAGENT includes a general system prompt, a research-topic-specific user prompt and history logs. The LLM generates multiple outputs, covering elements such as idea, methodology, DSL, etc. From these outputs, the one whose methodology has the greatest semantic difference from the previous round's methodology is selected as the idea for the current round, aiming to boost creativity in ideation.

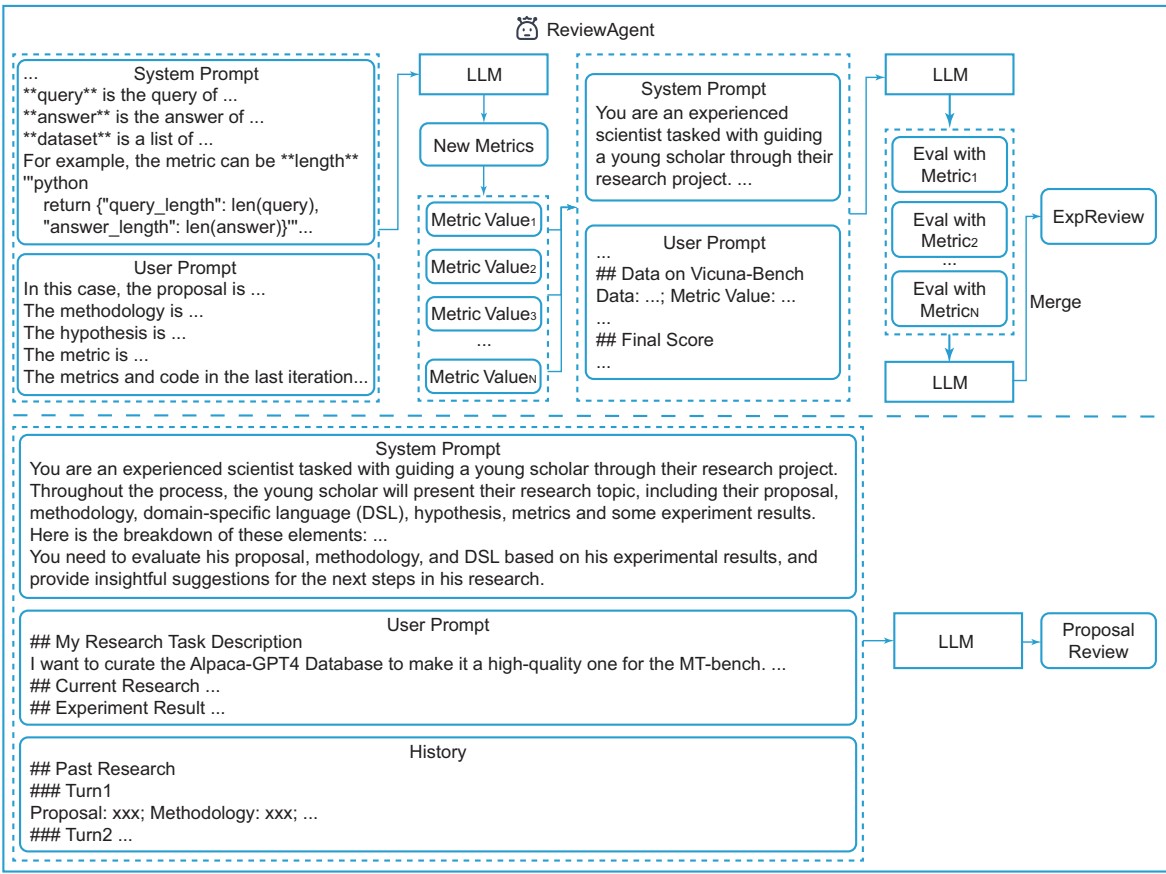

*Figure 12.* The REVIEWAGENT will first generate new metrics and then analyze each metric individually using the LLM. Following this, the REVIEWAGENT will call the LLM to merge the analysis results for each metric, resulting in the *ExpReview*. Next, the REVIEWAGENT will assess the experimental results by integrating insights from previous ideas and experiments, yielding the *ProposalReview*.

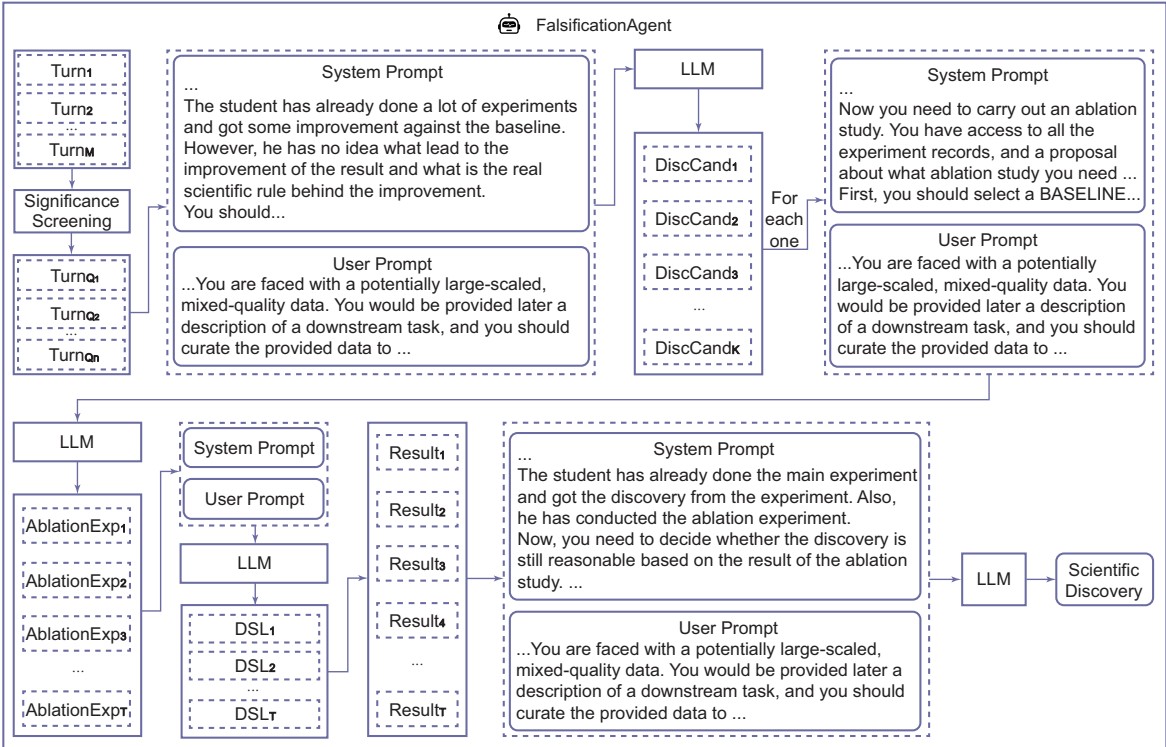

*Figure 13.* The FALSIFICATIONAGENT first screens all history turns to identify turns with notable changes in results. It then generates discovery candidates from the results obtained through significance screening. For each discovery candidate, it then creates several ablation experiment setups and generates the corresponding DSL to obtain experimental results. Once the experimental results are obtained, the FALSIFICATIONAGENT calls on the LLM to produce the final scientific discovery.

## D.4. Guidelines for Human Evaluators

To thoroughly assess the quality of our falsification process, we conducted a human evaluation of 20 agent-generated falsification logs. The guidelines are summarized as follows:

- ***Importance Score***: Assess the significance of the proposed scientific discovery candidate, considering its potential impact on experimental results and its relevance and consistency with the main experiments.

- ***Consistency Score***: Evaluate whether the proposed ablation experiments align with the scientific discovery candidate and whether the experiment appropriately isolates the factor in question.

- ***Correctness Score***: Determine whether the final scientific discovery drawn from the falsification process is correct based on the ablation and baseline results.

For each dimension, the evaluator assigns an integer score ranging from 0 to 2, where a higher score indicates better performance. The overall statistic results are shown in Table 2.

## D.5. Detailed Cases of BABY-AIGS Execution in Experiments

### D.5.1. DATA ENGINEERING

**PROPOSALAGENT**

---

**An example of the proposal from PROPOSALAGENT**

**Idea & Methodology**

**Idea**: ...Key issues identified include overly brief or excessively lengthy answers, lack of unique words, irrelevant content, poor adherence to instructions, lack of coherence, low keyword overlap, and poor sentiment balance...

- - - - - - - - - - - - - - - - - - - - - - - - - - - - - - - - - - - - - - - - - - - - - - - - - -

**Methodology**: **Key metrics to observe** include the *coherence of responses*, *adherence to instructions*, *relevance to the prompt*, *depth of information provided*, *clarity of instructions and responses*, *engagement in the conversation*...

**Experiment Settings**

Baseline: Iteration 0 *(the trivial method)*
Thought: ... we will filter the original dataset using the refined DSL with weighted criteria. ... and this will help in identifying the initial impact of the new criteria on the raw data and ensure that the dataset is not overly biased by similarity...

**Hypothesis & Related Feature**

**Hypothesis**: After using the processed data, the model's performance on the MT-bench task will **improve significantly**. The model should produce longer, more detailed, and coherent responses, ... The responses should be rich in unique words, and demonstrate appropriate sentiment balance compared to the baseline.

- - - - - - - - - - - - - - - - - - - - - - - - - - - - - - - - - - - - - - - - - - - - - - - - - -

**Related Feature**: ... length of responses, keyword overlap, unique word count, and sentiment balance.

**Rebuttal**

The review should provide an overall view of the experiment result, focusing on whether the selected examples effectively demonstrate improvements in the key metrics. The review should compare the performance of the model before and after the data curation to highlight the impact of the methodology. Specific examples should be used to illustrate both improvements and remaining issues to provide ...

---

**REVIEWAGENT**

---

**An example of the review from REVIEWAGENT**

**Review of the Experimental Results**

*Summary and Actionable Insights*: Based on the comprehensive analysis of various features influencing the scores of responses in the Alpaca-GPT4 Database, here are the key findings and recommendations for optimizing the dataset...
*Key Insights*:
1. Length and Word Count: High-quality responses tend to be longer, with word counts above 1000 for answers and around 15-20 words for queries.
2. Conciseness: While length...

**Review of the Proposal**

*Evaluation of Current Research Components*:
Your proposal effectively identifies key issues within the Alpaca-GPT4 dataset, such as... Additionally, the need for specific, measurable criteria for evaluating data points to improve...
*Suggestions*: 1. Data Distribution Analysis: Perform a quantitative analysis to understand the prevalence and distribution of these issues within your dataset...

---

**FALSIFICATIONAGENT** We present two examples of FALSIFICATIONAGENT: the first demonstrates the verification of a discovery candidate as correct, while the second illustrates the falsification of a discovery candidate.

---

### An example of the falsification process from FALSIFICATIONAGENT

#### Discovery Candidate

**Key Factor**: Importance of Context and Specificity.

#### Ablation Experiment Plan

Conduct an ablation study by systematically removing or altering one element related to context retention or specificity at a time. For example, test the impact of removing specific instructions or reducing context retention by limiting the number of conversational turns accessible to the model. This will help identify which specific factors within context and specificity contribute most significantly to model performance on MT-bench.

#### Methodology

**Methodology for Ablation Experiments**:{..."Principles": "...7. Responses should be concise and fall within the optimal length range (800-1500 characters).\n8. Responses should engage the user naturally and be informative.\n9. Weighting of each principle should be considered based on its importance to the downstream task.\n10. Incorporate dynamic thresholding to adjust based on the number of data points passing the initial filter."...}

- - - - - - - - - - - - - - - - - - - - - - - - - - - - - - - - - - - - - - - - - - - - - - -

**Methodology for Baseline Experiments**:{..."Principles": "...7. Responses should be concise and fall within the optimal length range (800-1500 characters).\n8. Responses should demonstrate context retention and follow multi-turn dialogue accurately.\n9. Responses should engage the user naturally and be informative.\n10. Weighting of each principle should be considered based on its importance to the downstream task.\n11. Incorporate dynamic thresholding to adjust based on the number of data points passing the initial filter.\n12. Break down complex criteria into more specific sub-criteria to capture nuances better. For example, 'context retention' can include sub-criteria like 'long-term memory' and 'contextual continuity.'\n13. Evaluate the impact of each criterion through ablation studies."...}

(**Highlights**: *The parts related to the candidate scientific factor are ablated from the baseline methodology (marked in red) to perform ablation experiments for verification.*)

#### Experiment Result

| Metric | Ablation | | Baseline | |
|---|---|---|---|---|
| | Trial 1 | Trial 2 | Trial 1 | Trial 2 |
| Vicuna-Bench (Validation) ↑ | 6.5500 | 6.7500 | 6.5250 | 6.5875 |
| MT-Bench (Test) ↓ | 4.0000 | 3.8313 | 3.9000 | 4.0750 |

#### Verification & Scientific Discovery

**Verification**: The core discovery highlights the importance of moderate context retention and strong logical progression for successful multi-turn dialogues. **In the first two experiments following the new PROPOSAL, the methodology likely focused on enhancing these aspects by using DSL to score and filter the data, potentially aiming for improved logical progression and context retention. The last two experiments with the baseline PROPOSAL probably did not emphasize these elements as strongly.** If the new PROPOSAL led to significant improvements in metrics relevant to multi-turn conversations (such as coherence and context relevance), then the discovery is indeed valuable.

- - - - - - - - - - - - - - - - - - - - - - - - - - - - - - - - - - - - - - - - - - - - - - -

**Scientific Discovery**: The true scientific discovery is that moderate context retention and strong logical progression are crucial for curating high-quality datasets for tasks requiring robust multi-turn dialogue capabilities.

---

**An example of the falsification process from FALSIFICATIONAGENT**

**Discovery Candidate**

**Key Factor**: Impact of Context Retention and Logical Progression.

**Ablation Experiment Plan**

Conduct an ablation study by systematically removing each criterion, starting with context retention and logical progression, and observe changes in model performance on the MT-bench. This will help identify the most impactful criteria and validate their importance in successful multi-turn dialogues.

**Methodology**

**Methodology for Ablation Experiments**:{..."Principles": "...7. Responses should be concise and fall within the optimal length range (800-1500 characters).\n8. Responses should engage the user naturally and be informative.\n9. Weighting of each principle should be considered based on its importance to the downstream task.\n10. Incorporate dynamic thresholding to adjust based on the number of data points passing the initial filter."...}

- - - - - - - - - - - - - - - - - - - - - - - - - - - - - - - - - - - - - - - - - - - - - - -

**Methodology for Baseline Experiments**:{..."Principles": "...7. Responses should be concise and fall within the optimal length range (800-1500 characters).\n8. Responses should demonstrate context retention and follow multi-turn dialogue accurately.\n9. Responses should engage the user naturally and be informative.\n10. Weighting of each principle should be considered based on its importance to the downstream task.\n11. Incorporate dynamic thresholding to adjust based on the number of data points passing the initial filter.\n12. Break down complex criteria into more specific sub-criteria to capture nuances better. For example, 'context retention' can include sub-criteria like 'long-term memory' and 'contextual continuity.'\n13. Evaluate the impact of each criterion through ablation studies."...}

(***Highlights***: *The parts related to the candidate scientific factor are ablated from the baseline methodology (marked in* red*) to perform ablation experiments for verification.*)

**Experiment Result**

| Metric | Ablation | | Baseline | |
|---|---|---|---|---|
| | Trial 1 | Trial 2 | Trial 1 | Trial 2 |
| Vicuna-Bench (Validation) ↑ | 7.1625 | 6.7500 | 6.475 | 6.5375 |
| MT-Bench (Test) ↑ | 4.10625 | 4.1125 | 4.05625 | 3.96875 |

**Verification & Scientific Discovery**

**Verification**: The discovery indicates that context retention and logical progression are important for the quality of multi-turn dialogues. However, the marginal improvements in scores suggest that these principles alone do not significantly enhance data quality. Thus, while they are important, they do not seem to drastically affect the data quality positively or negatively.

- - - - - - - - - - - - - - - - - - - - - - - - - - - - - - - - - - - - - - - - - - - - - - -

**Scientific Discovery**: The true scientific discovery is that while context retention and logical progression are important for multi-turn dialogues, their isolated application does not dramatically improve the dataset's quality for MT-bench. This suggests the need for a more nuanced and integrated approach, considering other quality metrics alongside these principles.

## D.5.2. SELF-INSTRUCT ALIGNMENT

**PROPOSALAGENT**

---

**An example of the proposal from PROPOSALAGENT**

**Idea & Methodology**

**Idea**: ...By generating diverse and challenging examples that address edge cases and complex scenarios, the goal is to enhance the dataset's relevance, specificity, and robustness...

- - - - - - - - - - - - - - - - - - - - - - - - - - - - - - - - - - - - - - - - - - - - - - -

**Methodology**: 1) Cross-Benchmark Alignment: ... 2) Adversarial and Synthetic Data Enhancement: ... 3) Contextual Relevance and Specificity: ... 4) Dynamic Feedback Incorporation: ... 5) Human-in-the-Loop Evaluation: ...

**Experiment Settings**

Baseline: Iteration 10
Thought: Enhance dataset quality by conducting cross-benchmark analysis to align adversarial and synthetic examples with MT-bench-specific requirements, and incorporate dynamic feedback for continuous refinement.

**Hypothesis & Related Feature**

**Hypothesis**: By applying the proposed methodology and utilizing the enhanced dataset ... Specifically, the model should exhibit higher accuracy, contextual relevance, and robustness in its responses, capable of handling a diverse range of instruction types and scenarios. The improvements in dataset quality and alignment with MT-bench-specific requirements should lead to more consistent performance gains across benchmarks ...

- - - - - - - - - - - - - - - - - - - - - - - - - - - - - - - - - - - - - - - - - - - - - - -

**Related Feature**: ... length, keyword overlap, instruction complexity, adversarial example ratio ...

---

**REVIEWAGENT**

---

**An example of the review from REVIEWAGENT**

**Review of the Experimental Results**

*Key Insights*:
1. Length and Balance: Ensure a moderate balance between query and answer lengths. Aim for query lengths between 44-122 characters and answer lengths between 940-3039 characters for optimal performance...
2. Keyword Overlap: Target a moderate keyword overlap of 3 to 7 between queries and answers... *Strategies*:
Data Rewriting and Augmentation: Use the insights from each feature to rewrite and augment your seed data. Focus on creating balanced, contextually relevant, and comprehensible instructions with appropriate keyword overlaps and redundancy levels ...

**Review of the Proposal**

*Evaluation of Current Research Components*:
... Despite the structured approach, the methodology may still lack a targeted focus on the specific nuances and requirements of MT-bench. Additionally, the feedback loop, while improved, could benefit from further refinement to ensure real-time adaptability and precise alignment with benchmark-specific criteria ...
*Suggestions*: 1. MT-bench-Specific Data Generation: ... 2. Bridge Dataset Development: ... 3. Dynamic Feedback Loop Enhancement: ... 4. Advanced Evaluation Metrics: ...

---

**FALSIFICATIONAGENT**   We present two examples of FALSIFICATIONAGENT: the first demonstrates the verification of a discovery candidate as correct, while the second illustrates the falsification of a discovery candidate.

---

### An example of the falsification process from FALSIFICATIONAGENT

**Discovery Candidate**

**Key Factor**: Importance of Contextual Relevance and Specificity.

**Ablation Experiment Plan**

Conduct an ablation study by removing enhancements specifically aimed at improving contextual relevance and specificity for MT-bench tasks. This includes eliminating the targeted adversarial and synthetic data generation techniques that were aligned with MT-bench-specific requirements. The goal is to assess if these targeted improvements contribute significantly to performance gains on MT-bench.

**Methodology**

**Methodology for Ablation Experiments**:{"thought": "Assess the impact of MT-bench-specific enhancements by removing them and observing changes in performance. This involves eliminating MT-bench-targeted adversarial and synthetic examples to understand their contribution to robustness and relevance.", "baseline": 13, "prompt": "Evaluate each instruction without focusing on MT-bench-specific enhancements. Remove targeted adversarial examples and synthetic variations that were specifically aligned with MT-bench. Observe performance changes to gauge the impact of these enhancements.", "seed": true}

**Experiment Result**

| Metric | Ablation | | Baseline | |
|---|---|---|---|---|
| | Trial 1 | Trial 2 | Trial 1 | Trial 2 |
| Vicuna-Bench (Validation) ↑ | 4.475 | 3.9875 | 4.15 | 4.6125 |
| MT-Bench (Test) ↑ | 2.7375 | 2.73125 | 2.75 | 2.886792 |

**Verification & Scientific Discovery**

**Verification**: The discovery still makes sense after the ablation study. The ablation experiments show that removing MT-bench-specific enhancements leads to a decrease in both validation and test scores compared to the baseline experiments. This supports the discovery that contextual relevance and specificity are important for enhancing model performance.

- - - - - - - - - - - - - - - - - - - - - - - - - - - - - - - - - - - - - - - - - - - - - - - - -

**Scientific Discovery**: The real scientific rule is that ensuring contextual relevance and specificity in instruction prompts contributes significantly to the performance of LLMs on instruction-following tasks. The improvements are notably due to the inclusion of targeted adversarial and synthetic data generation techniques aligned with specific benchmark requirements, like those of MT-bench.

---

**An example of the falsification process from FALSIFICATIONAGENT**

**Discovery Candidate**

**Key Factor**: Optimal instruction complexity.

**Ablation Experiment Plan**

To execute the ablation study, we will create subsets of the seed dataset with varying levels of instruction complexity: low, moderate, and high. Each subset will be used to train the model separately, and the performance will be evaluated on both VicunaBench and MT-bench. This will help us isolate instruction complexity as a variable and assess its impact on model performance across the benchmarks.

**Methodology**

**Methodology for Ablation Experiments**:{"thought": "Create subsets of the seed dataset by varying instruction complexity levels and analyze their impact on performance. Use these insights to refine dataset quality.", "baseline": 11, "prompt": "Divide the seed dataset into subsets with low, moderate, and high instruction complexity. Evaluate the LLM's performance on these subsets using VicunaBench and MT-bench benchmarks.", "seed": true}

**Experiment Result**

| Metric | Ablation | | Baseline | |
|---|---|---|---|---|
| | Trial 1 | Trial 2 | Trial 1 | Trial 2 |
| Vicuna-Bench (Validation) ↑ | 4.1375 | 4.725 | 4.3875 | 4.2625 |
| MT-Bench (Test) ↑ | 2.4625 | 2.975 | 3.175 | 2.73125 |

**Verification & Scientific Discovery**

**Verification**: The discovery of 'Optimal instruction complexity' doesn't hold consistently after the ablation study, as the results show varied performance. The ablation experiments indicate that moderate complexity instructions perform better on the validation dataset but inconsistently on the test dataset. The baseline experiments, focusing on cross-benchmark analysis and refinement, generally show higher and more stable scores across both validation and test datasets.

- - - - - - - - - - - - - - - - - - - - - - - - - - - - - - - - - - - - - - - - - -

**Scientific Discovery**: The real scientific rule appears to be that a comprehensive approach involving cross-benchmark analysis, dynamic feedback, and refinement (as seen in the baseline experiments) is more effective in improving LLM performance than simply focusing on instruction complexity. This suggests that a broader strategy that accounts for diverse data scenarios and continuous adaptation yields better results.

### D.5.3. LANGUAGE MODELING

**PROPOSALAGENT**

---

**An example of the proposal from PROPOSALAGENT**

**Idea & Methodology**

**Idea**: ... The current model configuration may not fully leverage diverse representations due to limited architectural features and static hyperparameters such as weight sharing and learning rate schedules ...

- - - - - - - - - - - - - - - - - - - - - - - - - - - - - - - - - - - - - - - - - - - - - - - - - - - - - - - - -

**Methodology**: 1) Adjust the number of attention heads ... 2) Experiment with lower dropout rates ... 3) Implement a more detailed cyclical learning rate schedule ... 4) Introduce weight decay ...

**Experiment Settings**

Baseline: Iteration 0 (the trivial method)
Thought: The current model may not be fully optimized due to suboptimal regularization and exploration of attention mechanisms. Additionally, the learning rate scheduling and optimization strategies could be further refined for better performance.

**Hypothesis & Related Feature**

**Hypothesis**: By refining the dropout rate, attention heads, learning rate schedule, and introducing weight decay, the validation loss and perplexity will decrease, indicating improved model performance ...

- - - - - - - - - - - - - - - - - - - - - - - - - - - - - - - - - - - - - - - - - - - - - - - - - - - - - - - - -

**Related Feature**: ... validation loss, perplexity, and learning rate dynamics ...

---

**REVIEWAGENT**

---

**An example of the review from REVIEWAGENT**

**Review of the Experimental Results**

*Key Insights*:
1. Deeper Analysis of Weight Sharing and Learning Rates: Investigate the specific impact of dynamic weight sharing mechanisms and cyclical learning rates. Consider conducting ablation studies to isolate these factors and understand their individual contributions ...
2. Explore Additional Architectural Variations: Experiment with different configurations of residual connections and layer normalization placements to find optimal arrangements ...

*Strategies*:
1) Hyperparameter Tuning: Conduct a thorough hyperparameter search to optimize learning rates, batch sizes, and other training parameters. 2) Larger Scale Experiments: If resources allow, extend the experiments to larger datasets or models to validate the scalability of your approach ...

**Review of the Proposal**

*Evaluation of Current Research Components*:
... The introduction of residual connections, layer normalization, dynamic weight sharing, and cyclical learning rates appears to have positively impacted the model's performance. The application of these techniques is a logical step based on the current understanding of deep learning models ...
*Suggestions*:
1. Regularization and Dropout: ... 2. Evaluation on Additional Metrics: ... 3. Data Augmentation Techniques: ...

---

**FALSIFICATIONAGENT**    We present two examples of FALSIFICATIONAGENT: the first demonstrates the verification of a discovery candidate as correct, while the second illustrates the falsification of a discovery candidate.

---

### An example of the falsification process from FALSIFICATIONAGENT

**Discovery Candidate**

**Key Factor**: Dropout Regularization

**Ablation Experiment Plan**

To investigate the individual impact of dropout regularization on model performance, we will conduct an ablation study by removing dropout. This will help us understand its contribution to the overall performance of the model. So We will conduct an experiment by setting the dropout rate to 0 in the model's parameters to see how this affects the training and validation performance, while keeping all other hyperparameters the same as in the baseline experiment.

**Methodology**

**Methodology for Ablation Experiments**:{"thought": "The aim is to assess the impact of dropout regularization by removing it completely and observing changes in model performance.", "advice": ["Set dropout rate to 0 to observe its impact on performance.", "Experiment with 12 attention heads to increase model expressiveness.", "Implement a cyclical learning rate with a shorter cycle length to allow more frequent adjustments.", "Introduce a weight decay of 0.05 to regularize the model."]}

**Experiment Result**

| Dataset | Trial | Baseline | Ablation |
|---|---|---|---|
| shakespeare_char | 1 | 1.495 | 1.541 |
| | 2 | 1.494 | 1.534 |
| | 3 | 1.509 | 1.543 |
| enwik8 | 1 | 0.984 | 0.997 |
| text8 | 1 | 0.966 | 0.967 |

**Verification & Scientific Discovery**

**Verification**: The discovery of the student still makes sense after the ablation study.

- - - - - - - - - - - - - - - - - - - - - - - - - - - - - - - - - - - - - - - - - - -

**Scientific Discovery**: The ablation study shows that dropout regularization plays a significant role in improving model performance. When the dropout rate is set to 0, there is a noticeable increase in loss across datasets, indicating that dropout contributes to better generalization by preventing overfitting. Weight decay also contributes to regularization, but the individual impact of dropout is crucial for achieving optimal performance.

## An example of the falsification process from FALSIFICATIONAGENT

### Discovery Candidate

**Key Factor**: the Number of Attention Heads

### Ablation Experiment Plan

To isolate the effect of increasing the number of attention heads from 8 to 12, we should revert the number of attention heads back to 8 while keeping all other modifications from Experiment 4. This will allow us to determine the impact of attention head expressiveness on model performance.

### Methodology

**Methodology for Ablation Experiments**:{"thought": "To isolate the effect of attention head count, we revert the attention heads to 8 while keeping other modifications constant from Experiment 4.", "advice": ["Reduce the dropout rate to 0.1 to see if it improves performance without overfitting.", "Keep attention heads constant at 8 to test the hypothesis.", "Implement a cyclical learning rate with a shorter cycle length to allow more frequent adjustments.", "Introduce a weight decay of 0.05 to regularize the model."]}

### Experiment Result

| Dataset | Trial | Baseline | Ablation |
|---|---|---|---|
| shakespeare_char | 1 | 1.495 | 1.483 |
| | 2 | 1.494 | 1.490 |
| | 3 | 1.509 | 1.481 |
| enwik8 | 1 | 0.984 | 1.000 |
| text8 | 1 | 0.966 | 0.972 |

### Verification & Scientific Discovery

**Verification**: The discovery of the student does not fully make sense after the ablation study. While the initial hypothesis was that increasing the number of attention heads improves model performance, the ablation study showed that even without increasing the attention heads, the model performance was similar or only slightly worse. This suggests that other factors, such as the reduced dropout rate, cyclical learning rate, and weight decay, might also significantly contribute to the performance gains.

- - - - - - - - - - - - - - - - - - - - - - - - - - - - - - - - - - - - - - - - - - - - - - - - - - - -

**Scientific Discovery**: The real scientific rule is that while increasing attention heads can enhance model expressiveness, other hyperparameter modifications such as dropout rate, learning rate scheduling, and weight decay play a crucial role in optimizing model performance. Therefore, attributing performance improvements solely to attention head count can be misleading without isolating its specific effects.

### D.5.4. DISCOVERYWORLD++

**Hypothesizer** We present the trajectory of Hypothesizer (Jansen et al., 2024) provided in the original DiscoveryWorld benchmark (Jansen et al., 2024) on our DiscoveryWorld++ environment.

---

**An example of Hypothesizer trajectory on Plant Nutrients task**

**Key Trajectories**

**Iteration 5:** Pick up the shovel.
**Iteration 7:** Pick up the soil nutrient meter.
**Iteration 9:** Pick up the seed jar.
**Iteration 11:** Use the soil nutrient meter on the soil of the Experimental Field.
**Iteration 12:** Use the shovel on the soil.
**Iteration 13:** Place a seed in the hole.
**Iteration 14:** Cover the seed with soil.
**Iteration 29:** Talk to the soil nutrient controller to set nutrient levels: Potassium low; Titanium low; Lithium low; Thorium low; Barium low.
**Iteration 75:** Talk to the soil nutrient controller to set nutrient levels: Potassium medium; Titanium medium; Lithium medium; Thorium medium; Barium medium.
**Iteration 112:** Use the soil nutrient meter on the soil of the Pilot Field.
**Iteration 227:** Talk to the soil nutrient controller to set nutrient levels: Potassium medium; Titanium medium; Lithium medium; Thorium medium; Barium medium.
**Iteration 248:** Submit answer.

**Statistical Result**

**Task Process:** 9 / 9     **Task Compeletion:** 1 / 1

**Ground Truth & Answer Submitted**

**Ground Truth**: Potassium with high level or Titanium with high level can both make the plant grow.

- - - - - - - - - - - - - - - - - - - - - - - - - - - - - - - - - - - - - - - - - - - - - - - - - - - -

**Answer Submitted**: Potassium at high levels and Barium at medium levels can promote plant growth.

## An example of Hypothesizer trajectory on Chemistry task

### Key Trajectories

**Iteration 1:** Pick up the rusted key.
**Iteration 5:** Pick up the jar.
**Iteration 7:** Use the dispenser labeled Substance A on the jar.
**Iteration 9:** Use the dispenser labeled Substance B on the jar.
**Iteration 11:** Use the dispenser labeled Substance C on the jar.
**Iteration 13:** Use the dispenser labeled Substance D on the jar.
**Iteration 14:** Place the rusted key into the jar containing the chemical mixture.
**Iteration 15:** Submit answer.

### Statistical Result

**Task Process:** 9 / 11     **Task Compeletion:** 0 / 1

### Ground Truth & Answer Submitted

**Ground Truth**: The rust remover compound is a mixture of exactly these compounds in exactly these proportions: 3 parts Substance B, and 1 part Substance C.

- - - - - - - - - - - - - - - - - - - - - - - - - - - - - - - - - - - - - - - - - - - - - - - - - - - - - -

**Answer Submitted**: The correct chemical combination to use to remove the rust from the key is a mixture of 1 part Substance A, 1 part Substance B, 1 part Substance C, and 1 part Substance D.

**FALSIFICATIONAGENT**    Due to the limited capabilities of current available LLMs to adapt to the complex environment of DiscoveryWorld++, we present a manually operated trajectory with explicit falsification process, which represents the FALSIFICATIONAGENT based on an ideal language model, to illustrate how falsification should be conducted on this modified environment.

---

### Manually operated process of falsification on Plant Nutrients task

#### Falsification for Potassium

**Explanation:** In order to falsify whether Potassium at high level can promote growth, we need to maintain other nutrients at the level that cannot promote growth and modify the amount level of Potassium.
**Action 1:** Set nutrient controller: Potassium high; Titanium low; Lithium low; Thorium low; Barium low.
**Observation 1:** The seed grows into a mushroom successfully.
**Action 2:** Set nutrient controller: Potassium low; Titanium low; Lithium low; Thorium low; Barium low.
**Observation 2:** The seed cannot grow.
**Conclusion:** The true discovery is: Potassium at high level can promote growth.

#### Falsification for Barium

**Explanation:** In order to falsify whether Barium at high level can promote growth, we need to maintain other nutrients at the level that cannot promote growth and modify the amount level of Barium.
**Action 1:** Set nutrient controller: Potassium low; Titanium low; Lithium low; Thorium low; Barium medium.
**Observation 1:** The seed cannot grow.
**Action 2:** Set nutrient controller: Potassium low; Titanium low; Lithium low; Thorium low; Barium high.
**Observation 2:** The seed cannot grow.
**Action 3:** Set nutrient controller: Potassium low; Titanium low; Lithium low; Thorium low; Barium low.
**Observation 3:** The seed cannot low.
**Conclusion:** The true discovery is: Barium at any level cannot promote growth.

#### Falsification for other nutrients

**Explanation:** As we have proved that Barium at any amount level cannot promote growth, we need to discover whether other nutrients can promote growth. We conduct experiments following the sequence: Titanium, Lithium and Thorium.
**Action 1:** Set nutrient controller: Potassium low; Titanium high; Lithium low; Thorium low; Barium low.
**Observation 1:** The seed grows into a mushroom successfully.
**Conclusion:** Titanium at high level can promote growth.
**Action 2:** Set nutrient controller: Potassium low; Titanium low; Lithium high; Thorium low; Barium low.
**Observation 2:** The seed cannot grow.
**Action 3:** Set nutrient controller: Potassium low; Titanium low; Lithium medium; Thorium low; Barium low.
**Observation 3:** The seed cannot low.
**Action 4:** Set nutrient controller: Potassium low; Titanium low; Lithium low; Thorium low; Barium low.
**Observation 4:** The seed cannot low.
**Conclusion:** Lithium at any level cannot promote growth.
**Action 5:** Set nutrient controller: Potassium low; Titanium low; Lithium low; Thorium high; Barium low.
**Observation 5:** The seed cannot grow.
**Action 6:** Set nutrient controller: Potassium low; Titanium low; Lithium low; Thorium medium; Barium low.
**Observation 6:** The seed cannot grow.
**Action 7:** Set nutrient controller: Potassium low; Titanium low; Lithium low; Thorium low; Barium low.
**Observation 7:** The seed cannot grow.
**Conclusion:** Thorium at any level cannot promote growth.

## Manually operated process of falsification on Chemistry task

### Figure out the mixture that can remove rust

**Explanation:** First, we need to get the mixture that can fully remove rust before conducting experiment for falsification.
**Action 1:** Use chemicals: 3 parts A, 3 parts B, 3 parts C and 3 parts D.
**Observation 1:** The rust is successfully removed
**Conclusion:** 3 parts A, 3 parts B, 3 parts C and 3 parts D is effectively for removing rust.

### Falsification for Substance A

**Explanation:** As rust has been removed, we can now start on falsification experiment. We first falsify what amount of Substance A is essential.
**Action 1:** Use chemicals: 2 parts A, 3 parts B, 3 parts C and 3 parts D.
**Observation 1:** The rust is successfully removed.
**Action 2:** Use chemicals: 1 part A, 3 parts B, 3 parts C and 3 parts D.
**Observation 2:** The rust is successfully removed.
**Action 3:** Use chemicals: 3 parts B, 3 parts C and 3 parts D.
**Observation 3:** The rust is successfully removed.
**Conclusion:** The true discovery is: Substance A is no use for removing rust.

### Falsification for Substance B

**Explanation:** Falsification on what amount of Substance B is essential.
**Action 1:** Use chemicals: 2 parts B, 3 parts C and 3 parts D.
**Observation 1:** The rust can not be removed.
**Conclusion:** The true discovery is: 3 parts of Substance B is essential for removing rust.

### Falsification for Substance C

**Explanation:** Falsification on what amount of Substance C is essential.
**Action 1:** Use chemicals: 3 parts B, 2 parts C and 3 parts D.
**Observation 1:** The rust is successfully removed.
**Action 2:** Use chemicals: 3 parts B, 1 part C and 3 parts D.
**Observation 2:** The rust is successfully removed.
**Action 3:** Use chemicals: 3 parts B and 3 parts D.
**Observation 3:** The rust can not be removed.
**Conclusion:** The true discovery is: 1 part of Substance C is essential for removing rust.

### Falsification for Substance D

**Explanation:** Falsification on what amount of Substance D is essential.
**Action 1:** Use chemicals: 3 parts B, 1 part C and 2 parts D.
**Observation 1:** The rust is successfully removed.
**Action 2:** Use chemicals: 3 parts B, 1 part C and 1 part D.
**Observation 2:** The rust is successfully removed.
**Action 3:** Use chemicals: 3 parts B and 1 part C.
**Observation 3:** The rust is successfully removed.
**Conclusion:** The true discovery is: Substance D is no use for removing rust.

