# OpenReview forum: "Position: Falsify, Don’t Just Discover - AI Generated Discoveries are NOT Born Scientific"
_ICML.cc/2025/Position_Paper_Track — Submitted to ICML 2025 Position Paper Track_

### Official Review · Reviewer_Ajqj · 2025-02-25

**Significance:** 2
**Argument Clarity:** 2
**Rating:** 2
**Confidence:** 4

**Questions:**

Based on the above, I would like a clarification on how the position expressed in this paper differs from the majority of works in the field, not from an abstract point of view of "falsifiability" but on whether the actual work done by the agent is significantly different. For example, the scores proposed by the authors can be applied to any experiment designed by the AI Scientist, in which case Tab. 2 would turn into a comparison of their framework with respect to BABY-AIGS.

**Discussion Potential:**

2

**Paper Summary:**

The paper considers the design of an AI agentic workflow for scientific discovery, where multiple LLM-based models cooperate to propose a novel hypothesis on a specific domain. The core position of the paper is that any workflow of this form should consider a "falsification" agent as the key component, in agreement with scientific falsifiability as proposed by Karl Popper.

Based on this proposal, the authors design a specific framework (called BABY-AIGS) as a proof-of-concept of the idea (see Fig. 3 for the workflow and Fig. 13 in the appendix for the "falsification agent"). The framework is run on three machine learning domains, so that the falsifiability step is implemented as specific ablation studies on the proposed algorithm.

They evaluate this workflow by designing three metric scores (with a focus on how good humans perceive the ablation study, and how accurate is the result in light of this ablation). The results show that the method significantly lags behind on the last metric despite proposing very promising ideas (see Tab. 2).

## Update after rebuttal

As per the discussion below, the paper's position is not "AI for science needs falsification", but "falsification in AI discovery should be given more weight / autonomy / prominence". All papers linked by the authors do have some form of "falsification", even if implicit (i.e., by tracking a given metric), but not necessarily a specific component or agent devoted to falsification. So, the score reflects mostly what we mean by "falsification" and whether the position provided here goes against the current consensus. I personally do not believe it to have much potential impact - I have increased a couple of scores but not the main one. However, I would be okay if the paper was eventually accepted in case all other reviewers disagreed on this point.

**Position:**

Yes

**Position In Title:**

Yes

**Related Work:**

3

**Strengths And Weaknesses:**

I will focus my review on the falsification agent, as (a) this is the position expressed in the paper, and (b) it is also the major technical innovation of BABY-AIGS.

My main concern is that I do not understand what is supposed to be the "alternative view" compared to this position. "Falsification" in the context of this paper means that an agent should propose explicit experiments that validate the method compared to specific baselines or ablations. But how are we supposed to design AI agents in alternative to this? Even if not explicit called "falsification", this step is present in almost every framework proposed up-to-date.

For example, in the AI Scientist by Sakana (https://arxiv.org/pdf/2408.06292) the agent in the first experiment proposes to "*Train models with both the original and new architecture on all datasets*" and "*Compare performance using KL divergence*". Isn't this an explicit falsification step? The authors argue in Section 4 (Alternative View) that these systems "could generate hypotheses anyway but falsify them largely by chance" without a falsification procedure. But the actual behavior of the two frameworks seems very similar to me, as eventually all ideas are subjected to ablations / baselines and compared on the basis of a chosen metric. Whether the agent explicitly has "falsifiability" in the name seems irrelevant to me.

In addition, the paper focus on scenarios with experiment-in-the-loop, such as machine learning. However, even in scenarios with in-vitro or social experiments (e.g., the AI co-scientist by Google, or the Virtual Lab) the result of the AI agent is a set of experiments that must be run and evaluated to assess the validity of the idea. Again, isn't this a potential falsification of the idea?

As a sidenote, I would like to note that the falsifiability theory has been subjected to a lot of scrutiny in the philosophy of science (e.g., does it hold in domains like string theory? are alternative explanations closer to the way science works in practice?). This is not discussed in this paper where the authors simply assume that the most basic form of falsifiability perfectly models "science". However, this is only true by restricting to the very specific domain of empirically-verifiable hypotheses subject to ablation studies which, ironically, makes alternative views to this position impossible (aka, unfalsifiable).

Finally, the paper has not undergone a strong proofreading (e.g., "the prosperity of autonomous agents", "they are refrained from AIGS").

**Support:**

2

---

> ### Author Rebuttal · Authors · 2025-04-01
>
> We appreciate the reviewer’s thoughtful concerns, and will clarify critical misconceptions and issues by points:
> ## Q1 (Position): Clarification on the position, proof-of-concept experiments, and relationship to prior works.
> First of all, we kindly request the reviewer to briefly read `W1` section in the rebuttal to Reviewer tepA.
>
> Next, we want to emphasize that our position is "**automated scientific discovery needs automated falsification**" (Abs. & Sec.1), instead of **proposing the falsification agent module in a multi-agent research system**. The former is a proposal for the research direction in a stage in AI-automated scientific discovery; and the falsification agent is a method design of the proof-of-concept experiment.
>
> That is, we are not proposing plugging in the falsification agent for other works, but advocating others to explicitly study more on the falsification stage of their systems, regarding, e.g., the efficiency of falsification to identify scientific discoveries, the quality of ablation study, the correctness of experimental results interpretation, (L307 Sec.3.2) and etc. In summary, **AI-automated research systems should treat the falsification stage as an explicit subject to design, optimize, and evaluate**, as a few follow-up studies [1].
>
> Importantly, we want to clarify that ideal falsification should not only be "validating compared to baselines or ablations", but to proactively and correctly come up and refute any possible alternatives that could contradict the hypothesis (L63 Sec.1, L250 Sec.3.1, L284 Sec.3.2), e.g. heliocentric to geocentric theory (Fig.1).
>
> We could then adress the reviewer's issue about AI Scientist. It is a pioneer work to generate AI-automated research papers. Could it do falsification? Yes, as the papers are with empirical analysis and sometimes with extensive ablation study. Is the system the ideal format in terms of falsification? We are afraid not. AI Scientist (up to submission date) still implements falsification implicitly in terms of structure design, and still faces instability in creativity and executability according to our experiments, which are critical principles for advancing falsification capability (Sec.3.2). **We believe that further optimizing the falsification process explicitly in prior works, including AI Scientist, is helpful and necessary**.
>
> The misinterpretation incorrectly assumes our analysis (Table 1, Sec.4) on AI Scientist, whereas **our comparison is with the ideal AIGS system, not Baby-AIGS**. We argue that merely allowing explicit falsification steps to happen is insufficient without systematic studies, as AI Scientist cannot guarantee effective falsification—due, for instance, to incapable LLMs or failed ablation experiments. Notably, enhancing falsification abilities in the backbone model also aligns with our emphasis on explicitness.
>
> We hope these could address the concern of the reviewer. And we will carefully incorporate the clarification in the revised manuscript.
> ## W1 (Generalization): Could falsification be applied to non-empirical subjects?
> We thank the reviewer for highlighting in-vitro and social applications for automated falsification. We clarify that AI Co-Scientist was published after our submission, and Virtual Lab is discussed in L972 App.A. Both works belong to the research copilot category (Sec.2.2): AI Co-Scientist generates proposals with scaled test-time compute, and Virtual Lab generates protein designs. Their falsification experiments are explicitly human-executed, contradicting our goal of automation. We suggest comparing these works to Laboratory Mobile Robots (Table 1), which explicitly falsify synthetic chemicals robotically but cannot propose hypotheses, also failing to fully automate science. Nonetheless, we plan expansions of Baby-AIGS in future due to limited resources (App.A).
>
> For the philosophy of science issue, we have explained thoroughly in Sec.3.1 that the falsification could be performed in either theoretical or empirical ways, and **we enumerate physics, mathematics, humanities, etc., to explain how theoretical findings could lead to falsification in the broader field of research**, including string theories. We limit the scope in ML only to ensure the completeness of our study, which should not be viewed negative.
> ## Q2 (Proof-of-Concept Experiment): What does Table 2 indicate? (Sec.5.3)
> Please refer to `W1` to Reviewer tepA. The conclusions support our position and we do not think comparing to AI Scientist will be further beneficial. Ideally, falsification should be performed in or over human levels, but not for current Baby-AIGS **in this position paper**.
> ## W2 (Presentation)
> Thanks for pointing out the language issue. We will clarify and improve the presentation in the next version of manuscript.
>
> ---
> **Thank you for your time and for considering these clarifications.**
>
> [1] Huang et al. Automated Hypothesis Validation with Agentic Sequential Falsifications. Feb 14, 2025.

---

> > ### Comment · Reviewer_Ajqj · 2025-04-02
> >
> > I thank the authors for the detailed response. I agree that performing ablations or comparing on a pre-specified metric is a limited form of falsification, and other possibilities are available. I also agree that future works should stress more this component in the design of the agentic systems. However, I also remain convinced that, in the limited setup considered by the authors where falsification is possible "in-the-loop", the use of ablations and metrics is proof that "falsification" is the only viable option, and no clear alternatives exist. Thus, whether we need "more falsification" is a position with limited impact, as I think this is the shared consensus - and the paper becomes more a way to advertise the authors' own work. Nonetheless, I have updated my score to a weak reject, and I remain open to a further re-evaluation based on the reviewers' discussion.

---

> > > ### Author Response · Authors · 2025-04-09
> > >
> > > We thank the reviewer for the timely response, allowing us to further address the concerns beyond the context limits in the first rebuttal.
> > > ## W1 (Position): Clarification on the position and settings.
> > > Firstly, we would like to clarify our primary position: "**Automated scientific discovery necessitates automated falsification**". Our argument emphasizes the "**automation**" of falsification as an enabler of science automation.
> > >
> > > For our paper's setting, we address "**empirical subjects that requires actual implementation of the methodology to obtain empirical results for falsification, e.g., machine learning**" (Sec. 3.1). We fully agree that falsification, well-established since Popper (1935), remains fundamental in empirical science. **However, we specifically advocates to increase autonomy within the falsification process**. Thus, rather than restricting our setting, the established importance of falsification actually reinforces our call for autonomous falsification.
> > >
> > > Furthermore, **the autonomy necessitates certain key requirements**. An ideal system should explicitly design and optimize the falsification process, conduct with minimal human intervention, and employ scalable approaches (Table 1). In Sec. 3.2, we highlight two critical principles essential for such a system: *executability and creativity*. **Present research have yet to fully resolve these challenges or demonstrated a clear favor to prioritize them.** Limitations include dependence on human intervention (e.g., AlphaFold, Virtual Lab), constrained creativity (e.g., Lab-Robots), inconsistent performance in implicit falsification methods (e.g., AI Scientist, AI Hilbert), and restricted domain generalization (e.g., DataVoyager). Recent studies [1,2] have begun structurally optimizing falsification processes, progressively advancing toward greater autonomy. Nevertheless, further work is necessary to realize a fully autonomous AIGS system.
> > > ## Q1 (Alternative Views): What views are opposing "automated falsification" for science automation?
> > > The first category of viewpoints contradicts the automation. As depicted in Para.1, Sec.4, long reasoning systems, AI4Science models, and etc. achieve high performance in specific domains. However, when facing frontier problems, the solution summoned by these systems still needs to be manually falsified. AI Co-Scientist and Virtual Lab also lie in this category. **This viewpoint is not minor in academia**, e.g., Reviewer tepA.
> > >
> > > The second group assigns falsification lower priority. As noted in Para.2, Sec.4, falsification becomes unnecessary when only products, not scientific discoveries, are needed. For instance, robots can successfully synthesize desired chemicals through trials without generating meaningful discoveries. AI Scientist represents the opposition, where ideation and experimentation are rigorously designed, yet with no explicit module for ablations. Consequently, while great works [3] with detailed falsifications exist, many lower-quality results still produce invalid or coarse falsifications. **This highlights the need for further improvements beyond existing considerations**.
> > >
> > > The third group think falsification autonomy does not require much effort and could already be implemented. For closed datasets, statistical approaches can refute hypotheses which are mathematically represented. **This group limits generalization and only fulfills automated falsification in a fixed dataset**.
> > >
> > > We will clarify these statements in the revised manuscript.
> > > ## W2 (Impact Statement and Self-Advertisement): Is "more" investigation in falsification autonomy a position with minor impact?
> > > As discussed above, there are series of works neglecting effort in falsification autonomy. And we argue that **a position suggesting investigating "more" in a research direction does not inherently bring a minor impact**. It is critical to point out the challenges (L307), the research principles (L325), and the potential that current AI systems could make valid falsification as human experts (Sec.5.3). Beyond, we provide metrics for falsification (Sec.5.2), and trajectory examples showing automated falsification improves AI-generated discoveries (L426). We also discuss the potential negative impact, and emphasize the benefits to ethical development (L440).
> > >
> > > For self-advertisement, we apologize for the negative impression we may have given; it was not our intention. We thoroughly discussed existing studies in Sec.2 and remained Baby-AIGS only in the proof-of-concept experiment. **We sincerely hope—and trust—that this unrelated factor will not influence the impartiality of review standards, which should focus on the support and significance of the position itself.**
> > >
> > > ---
> > > **Thank you again for your time and for considering these clarifications.**
> > >
> > > [2] Jansen et al. CodeScientist: End-to-End Semi-Automated Scientific Discovery with Code-based Experimentation. Mar 20, 2025.
> > > [3] https://sakana.ai/ai-scientist-first-publication/

---

### Official Review · Reviewer_xYAM · 2025-03-01

**Significance:** 3
**Argument Clarity:** 2
**Rating:** 3
**Confidence:** 4

**Questions:**

* Can you describe why you thought conducting the experiments in Section 5 was necessary? It seems like there could be other ways to support the utility of Baby-AIGS.
* Can you provide a set of research directions that you think would be valuable for the machine learning community to engage on?
* Are there limitations of the falsification approach that require research in additional dimensions?

**Discussion Potential:**

3

**Paper Summary:**

The paper argues that AI-enabled, automated scientific discovery needs automated falsification of proposed hypotheses. The position draws from prior arguments that stated that falsification is the central component of scientific research and as such should be incorporated in AI-based systems for scientific discovery. Section 1 outlines an introduction to previous efforts in AI for Science and introduces the paradigm of falsification in the context of AI-agents named AI-Generated Science (AIGS). After introducing AIGS the paper divides current AI for science research efforts into three categories: 1. frameworks for performance optimization; 2. automatically proposing scientific hypotheses; 3. automated scientific discovery. The introduction finishes by introducing Baby-AIGS, which the authors describe as the initial "baby" step towards autonomous discovery systems that propose ideas and incorporate the process of falsification explicitly.

Section 2 provides a discussion related to the development of AI-Accelerated scientific discovery divided into four categories: 1. AI as a performance optimizer for concrete machine learning based tasks; 2. AI as a research assistant providing partial automation for common scientific tasks such as literature review; 3. AI as an automated scientist with the ability to execute the full scientific process; 4. AI-based research communities that foster collaboration between scientific agents. Section 3 details a deeper discussion on the role of falsification in the scientific process, which includes a definition and a brief summary of how prior efforts have not fully taken falsification into account. The paper also proposes a new framework for incorporating falsification into automated systems for scientific discovery based on a pre-falsification stage that focuses on generating hypotheses and a falsification stage that designs experiments to probe those hypotheses.

Section 4 details a brief discussion of alternative views mostly focusing on how previous efforts (e.g., performance optimizers and task automation) fall short in achieving full AIGS. Section 5 describes the Baby-AIGS system that explicitly incorporating a pre-falsification stage and a falsification stage. Next, the paper describes an experiments based on the Baby-AIGS system that include open-world and closed-world experiments focusing on machine learning and scientific discovery research challenges. The experiment includes an evaluation of falsification based on human feedback along three dimensions (importance, consistency, correctness) with an overall score. Based on the experimental results, the authors make the conclusions stating that Bay-AIGS can create valid scientific discoveries but lags behind human researchers. A third conclusion claims that falsification is the missing link for making AI-generated scientific discoveries more reliably scientific. Before the conclusion, the paper discusses how current benchmark fail to explicitly include falsification and outlines supplementary evaluations along falsification, creativity and executability.

## update after rebuttal

I appreciate the authors' additional details and clarification and remain supportive of the paper assuming the final version includes the additional clarity outlined in the discussion period. Overall, I maintain that the paper provides useful points of discussion for the community along with good support for its central claims. I think it may also be worth emphasizing that the experiments presented may serve as a starting point for future research.

**Position:**

Yes

**Position In Title:**

Yes

**Related Work:**

2

**Strengths And Weaknesses:**

Overall, the paper provides an interesting and potentially valuable position on the importance of falsification in AI-based scientific that could lead to engaging discussions related to design principles of automated scientific AI systems. That being said, the current draft of the paper could be significantly improved by providing greater clarity and focus for the core position and adding additional context from different scientific fields.

**Strengths:**
* The paper's position is interesting and has viable potential for discussion to inform future research directions.
* The paper supports its position based on the importance of falsification in the scientific process in general and outlines relevant context for how previous AI for science research falls short in falsification. The classification and description of previous AI for Science efforts is also useful as a framework.

**Weaknesses:**
* The paper's clarity could be significantly improved. The paper includes many different ideas in different sections, some of which are described in isolation without clear links to the main position. For example, the "AI forms a research community" section does not have a clear link to the main position and is not discussed beyond Section 2.
* While the proposal of the Baby-AIGS framework is useful, the purpose of the experiments in Section 5 is unclear. Much of the relevant details related to the claims in Section 5 are in the appendix and the analysis seems to involve both agents and human evaluators. The paper would be substantially involved with greater clarity.
* The alternative views section could be further improved with a discussion on why falsification may not be important to the scientific process and what alternative framings could exist. In it's current form, Section 4 mainly repeats the claims from previous sections.
* The paper could be further improved by discussing the development of different agent systems in scientific fields [1] [2] beyond the current discussion of open and closed world environments. This can be further improved by incorporating other frameworks for scientifically informed discovery [3].

References

[1] Huan Zhang, Yu Song, Ziyu Hou, Santiago Miret, and Bang Liu. 2024. HoneyComb: A Flexible LLM-Based Agent System for Materials Science. In Findings of the Association for Computational Linguistics: EMNLP 2024, pages 3369–3382, Miami, Florida, USA. Association for Computational Linguistics.

[2] Ramos MC, Collison CJ, White AD. A review of large language models and autonomous agents in chemistry. Chemical Science. 2025.

[3] Miret, S. and Krishnan, N.M., 2024. Are LLMs ready for real-world materials discovery?. arXiv preprint arXiv:2402.05200.

**Support:**

3

---

> ### Author Rebuttal · Authors · 2025-04-01
>
> Thank you for your valuable review and detailed comments. We sincerely appreciate your insightful suggestions and will address your concerns point-by-point below:
>
> ## Q1 & W2 (Experiments): Clarify necessity, purpose, and details of experiments in Section 5.
>
> We conducted the experiments in Section 5 to empirically validate the feasibility and necessity of automated falsification within the Baby-AIGS framework. The Baby-AIGS system is fully autonomous without human intervention, though we used human evaluators for rigorous falsification assessment.
>
> The primary goal of these experiments was twofold: first, to show that automated falsification is currently feasible and contributes to scientifically rigorous discoveries; second, to highlight the limitations in the current absolute performance level, reinforcing our position that automated falsification deserves greater attention and further research. Due to page constraints, details on creativity, executability, and additional closed-world experiments were placed in the appendix to maintain completeness and transparency.
>
> ## Q2 (Research Directions): Suggest valuable future ML research directions.
>
> The nascent AIGS paradigm presents several promising research directions. Our paper emphasizes the critical role of falsification. In Appendix A, we already outlined actionable insights and directions we consider valuable for further exploration, including:
> - Balance idea diversity and system executability.
> - Establish systematic mechanisms for evaluation and feedback.
> - Strengthen the falsification procedure.
> - Expand channels for scientific knowledge dissemination.
> - Exploring communication dynamics among autonomous AI researchers.
> - Promote interdisciplinary knowledge integration and experimentation.
>
> We will succinctly highlight these directions in the main text to increase visibility.
>
> ## Q3 (Falsification Limitations): Address limitations and additional research dimensions needed for falsification.
>
> We believe the greatest challenge in practicing falsification arises from the varying forms and degrees of falsification required across different scientific disciplines. In this paper, through the Baby-AIGS system, we demonstrate the feasibility and necessity of falsification specifically within the domain of machine learning. When extending such practices to other scientific domains, understanding and integrating domain-specific knowledge becomes even more crucial and challenging. We elaborate on this perspective further in Appendix A.
>
> ## W1 (Clarity): Improve clarity by better linking isolated sections to the central position.
>
> Thank you for highlighting the clarity issue. We recognize that the concept of AIGS indeed involves substantial ambiguity that requires detailed definition. For example, clarifying what types of results constitute genuine scientific discoveries and determining precisely which processes qualify as AI-generated. Thus, we provided a comprehensive review of AI-Accelerated Scientific Discovery to help readers understand recent developments in Section 2. Regarding your point about the section "AI forms a research community," at the time of our paper submission, there were no existing works implementing such a community. We are excited to observe recent preliminary attempts testing this view:
> - Schmidgall S, Moor M. AgentRxiv: Towards Collaborative Autonomous Research[J]. arXiv preprint arXiv:2503.18102, 2025.
>
> We will further include recent works that validate and support our perspective in  our revised manuscript.
>
> ## W3 (Alternative Views): Enhance discussion on cases where falsification might be unnecessary and alternative framings.
>
> As discussed in Table 1 and Section 4, we categorized representative AI-powered scientific discovery works into three types: evolving performance optimizers in specific domains, AI-empowered ideation and experimentation, and AI-empowered data interpreters. In the Section 4, we provided a detailed analysis explaining why these existing approaches fall short of realizing full-process automated AIGS systems compared to the ideal AIGS system. We welcome further discussion on these alternative views and appreciate your suggestions.
>
> ## W4 (Agent Systems): Broaden discussion with additional agent systems and scientifically informed frameworks.
>
> Indeed, our intention in Section 2 was precisely to discuss the development of agent systems across different scientific domains and paradigms. We categorized existing works into three lines of research: AI as a Performance Optimizer, AI as a Research Assistant, and AI as an Automated Scientist, noting that none of these existing frameworks explicitly integrated falsification.
>
> The distinction between open-world and closed-world experiments was deliberate, designed as a proof-of-concept to examine the feasibility of automated falsification in varied environments. We appreciate your recommendation and will incorporate suggested literature to enrich our discussion.

---

> > ### Comment · Reviewer_xYAM · 2025-04-03
> >
> > Thank you for the additional details. In terms of cases where falsification might be unnecessary, it might be useful to discuss how falsification interact with the mechanism of new discoveries that are confirmed by experiment. For example: a new equation that governs physical behavior is confirmed by observables; a AI-discovered protein is confirmed to have a particularly useful property for biotechnology. While this is partially covered by ideation and experimentation, it would be good to have more details on how falsification interacts or doesn't, is or is not needed, in these scenarios.

---

> > > ### Author Response · Authors · 2025-04-06
> > >
> > > We sincerely thank you for your thoughtful follow-up.
> > >
> > > In Section 3.1 of our paper, we discuss thoroughly that **the falsification could be performed in either theoretical or empirical ways**, and we enumerate physics, mathematics, humanities, etc., to explain how theoretical findings could lead to falsification in the broader field of research. To reiterate and expand on that point, we clarify that falsification does not merely apply to scenarios involving explicit negation of hypotheses through failed experiments; rather, it fundamentally includes the rigorous process of hypothesis verification via empirical or theoretical means. Specifically, a scientific discovery arises when a hypothesis withstands attempts at falsification and thus gains validation through rigorous testing.
> > >
> > > **Addressing the reviewer's examples:** A newly proposed equation that accurately describes observed physical behaviors or an AI-discovered protein validated to possess desirable biotechnological properties are both inherently subject to falsification. For instance, the new equation's predictions could be empirically tested against observational data, attempting to falsify its predictions. Similarly, the functionality of an AI-discovered protein can be rigorously evaluated through laboratory experiments to verify its properties. In both scenarios, falsification acts as the essential process confirming these hypotheses as genuine scientific discoveries rather than random or incidental findings. As discussed in Section 3.1 (lines 242–250), we consider these examples part of the falsification process; specifically, as you noted, "is confirmed by observables" and "is confirmed to have a particularly useful property for biotechnology" **are instances of falsification.**
> > >
> > > However, as discussed in Section 4 and illustrated in Table 1, current works in AI-accelerated scientific discovery fail to achieve full-process automated AIGS systems. This is primarily due to the absence of an explicit falsification process, as well as the difficulty of realizing fully autonomous and scalable optimization and falsification loops, thereby failing to autonomously generate generalized scientific discoveries. As outlined in Appendix A, we will continue to advance our efforts toward building ideal AIGS systems.
> > >
> > > We hope this clarifies our perspective: **automated falsification** is a critical and frequently underemphasized element of **AI-automated scientific discovery research, and it plays an indispensable role in moving toward fully automated science.** We welcome any further discussion or requests for clarification you may have.

---

### Official Review · Reviewer_NiCa · 2025-03-14

**Significance:** 4
**Argument Clarity:** 4
**Rating:** 5
**Confidence:** 4

**Questions:**

How well does this generalize from ablation studies. What are the adjustments/limitation is other hypothesis-driven scientific discovery.
How could traditional algorithms be used to more efficiently/intelligently prune the search space of possible perturbations (which ablations/how much/etc)

**Discussion Potential:**

3

**Paper Summary:**

This paper proposes a methodology focused on a purposeful methodology of falsification of hypothesis-driven AI generated science inline with Karl Popper theories. It discusses the theoretical importance of this overlooked approach and explores the value of this approach with expert human evaluators to assess the current performance and weaknesses of such a stratagem using ablation studies.

**Position:**

Yes

**Position In Title:**

Yes

**Related Work:**

3

**Strengths And Weaknesses:**

Strong in theory, background, argument, design, and validation.
Would benefit from greater discussion on limitations to generalizing to other hypothesis-driven scientific discovery beyond ablations. Perhaps create a taxonomy of different approaches and how this methodology is applicable (e.g.

**Support:**

4

---

> ### Author Rebuttal · Authors · 2025-04-01
>
> Thank you for your supportive assessments and insightful questions! We would like to answer your questions as follows.
>
> ## Q1(Generalization): How well does this generalize from ablation studies
> Ablation studies, which systematically remove or modify components of a system to evaluate their impact, are a specific experimental approach often used in research to validate performance or robustness. While our Baby-AIGS system mainly performs ablation experiments to falsify research hypotheses, the principle of falsification we advocate—designing experiments to refute hypotheses—generalizes beyond this context. There are other approaches for automated falsification other than ablations. For instance, the system can generate and test hypotheses by systematically perturbing variables or conditions, conduct related experiments to find more evidence or verify the target hypothesis by designing complementary experiments to falsify the opposite. The notion of falsification can generalize to different scenarios and implementations. We choose ablation studies since they are typical and widely-used in ML research areas.
>
> ## Q2(Limitations): What are the adjustments/limitation is other hypothesis-driven scientific discovery
> Our paper emphasizes that automated falsification is a missing yet essential component for fully realizing automated scientific discovery. Traditional hypothesis-driven scientific discovery methods mainly focus on extracting patterns from collected data. Their hypothesis validation process is not directly applicable for end-to-end scientific research, in which verifying research hypotheses require comprehensive experiment design, robust experiment execution and reflection on the results. As for the recently risen automated LLM-based research system, they basically lack a well-designed explicit falsification process, leading to unreliable research results.
>
> ## Q3(Pruning) How could traditional algorithms be used to more efficiently/intelligently prune the search space of possible perturbations
> In the context of AIGS and automated falsification, pruning the search space of possible perturbations—i.e., the range of experimental conditions or variables tested to refute a hypothesis — is critical for efficiency and scalability. We agree that traditional algorithms, like Monte Carlo search, Decision Trees or Rule-Based Systems, may complement the capabilities of LLMs and autonomous agents in this regard, offering structured and computationally efficient approaches to enhance the intelligence of the falsification process. We appreciate your insightful suggestions and will discuss its potential in our paper.

---

### Official Review · Reviewer_tepA · 2025-03-15

**Significance:** 1
**Argument Clarity:** 3
**Rating:** 1
**Confidence:** 3

**Questions:**

Do you really think that it would be "science" for a group of AI agents to do science, talking only to each other (and, according to the infographic, not looking at data)? Aren't humans important to this? Indeed the paper seems to imply that humans would have written these AI agents. If it is all AI, why don't we just leave it to them?

**Discussion Potential:**

4

**Paper Summary:**

The paper argues that ML methods should have a concept of falsification, and they should employ it as they do scientific tasks.

## update after rebuttal
My disagreements with the authors about the relevance of the scenario (which seems highly futuristic to me) remain. I note that position papers shouldn't require reviewer unanimity to be published!

**Position:**

Yes

**Position In Title:**

No

**Related Work:**

1

**Strengths And Weaknesses:**

Strengths:

Falsification is indeed an important idea in the execution of science.

Weaknesses:

The paper is based on a futuristic model in which AI agents are *doing science*. I have two problems with this. The first is that we are *very far* from this situation. The second is that *machines doing science isn't, itself, science*. Science is about human understanding, not robot understanding.

Right now, ML in science *is* used to falsify scientific hypotheses. Literally every paper I have ever written using ML in physics has been using ML to build a model for falsification. I am marking this as "poor" on prior work because I think (almost) every use of AI in science is about falsification. For example, the DeepMind work finding crystal structures was about *falsifying* bad crystal structures, and so on and so on.

The title only implies the position; it does not state it. Someone reading this paper could believe that the falsification should be applied *to* the ML models (indeed that's what I expected). Actually the proposal is that falsification should be applied *by* the ML models.

Paper had a flavor of being an advertisement for a particular AI effort. I didn't read the appendices, but I noticed that it had a huge proposal-like appendix.

**Support:**

2

---

> ### Author Rebuttal · Authors · 2025-04-01
>
> We thank the reviewer for the comments but respectfully address some critical misunderstandings by points:
>
> ## Q1 (Futuristics): Are we far from AI agents automating science?
> We acknowledge that today’s AI systems are still in relatively early stages of **fully** autonomous science. However, this area has become popular. (1) We provide a series of AI-automated research systems in Sec.2.3. Despite their limitations (Sec.4), it is clear that **several well-defined, simple research problems could be automated by current AI**. (2) A few follow-up works [1,2,3,4] reveal the validity for AI agents to perform full-process research and even produce conference (workshop)-level papers.
>
> Since the emergence of automated AI research systems has shown, we argue that **(1) more and more tasks and domains will involve automated research in the foreseeable future, and (2) we must start discussing the expected format of these systems, of which automated falsification is an important part**.
> ## Q2 (Background): Is science discovering subjective to AI and humans?
> This question is about philosophy of science. Our work builds upon Popper's definition of science based on falsification (Sec.3), which is publicly accepted as the mainstream and does not deny the possibility for AI to discover science. We reviewed a series of literature (Sec.2.3) showing part of academia reached consensus that AI could discover science as humans, at least in specific domains. AI-generated works [2,3] even show the possibility to undergo peer review from humans. **We barely see evidence or rationale for labeling a discovery as "scientific" when made by humans but withholding that label when the identical discovery is made by AI**.
> ## W1 (Position): Clarification on the position, prior works, and proof-of-concept experiments.
> We clarify our position (bold in the abstract and Sec.1): **automated scientific discovery needs automated falsification**. This differs from applying falsification via ML models in three ways: (1) **Scenario**: our position targets AI conducting end-to-end research autonomously, not aiding human research. (2) **Scale**: we focus on AI as automated scientists with general research capabilities (Sec.2.3), not ML models tackling specific tasks. (3) **Approach**: current AI (e.g., AlphaFold) serves as optimizers for particular problems (Sec.2.1), with falsification still reliant on humans, limiting full automation (Sec.4, Table 1).
>
> To support our position, we systematically review literature (Sec.2) to show the emergence of AI-automated research systems and the lack of systematic studies explicitly in falsification. We then design a proof-of-concept experiment to show that (1) automated falsification is needed in addition to AI discovery, (2) current AI could perform valid falsification with space to improve, and (3) current agent benchmarks lack evaluating falsification.
>
> Thus, automated falsification should be studied extensively towards full-process AI-automated scientific discovery.
> ## Q3 (Ideal Methodology): Could science be automated with a group of AI agents?
> The Baby-AIGS system is the proof-of-concept method for science automation. We use the agent system to support that falsification is important and should be studied more extensively, instead of claiming it as the ideal format to achieve science automation. **It is also incorrect that the system does not access data**. The experiment agent and review agent rigorously collect and analyze empirical data from real experiments.
>
> Our vision is that the science could be eventually automated for a large portion of areas compared to today's research. However, the multi-agent system could be a temporal format adopted widely (also in works in Sec.2.3), or the ultimate format with more development. However, **our position still holds that we do not think science could be automated without automated falsification**.
> ## Q4 (Broader Impact): What is the format of human research when AI could automate science?
> Notably, **we did not advocate whether humans are important in science discovering or whether falsification is important in human research, which is beyond the scope of this study**. Although the rise of AI-automated science is inevitable, we prioritize human benefit and ethical considerations. Advancing AI-driven automation in science need not harm humans; instead, it could accelerate innovation if properly regulated. At an early age, humans could use more meta-creativity in developing better AI researchers, with automated falsification for regulation (Impact Statement); and eventually, we personally believe and eager to leave most research problems to trust-worthy AI.
>
> ---
> **Thank you for your time and for considering these clarifications.**
>
> [1] Schmidgall et al. AgentRxiv: Towards Collaborative Autonomous Research. 2025.
> [2] https://sakana.ai/ai-scientist-first-publication/
> [3] https://www.intology.ai/blog/zochi-tech-report
> [4] https://github.com/HKUDS/AI-Researcher

---

> > ### Comment · Reviewer_tepA · 2025-04-05
> >
> > I thank the authors for taking this seriously. However, my disagreements remain, and I maintain my scores. I remind the Area Chairs that I don't feel exceedingly confident on this, especially since the authors and I disagree on what constitutes "science"!

---

### Decision · Program_Chairs · 2025-04-27

**Decision:**

Reject

**Comment:**

This paper received divergent reviews. While two reviewers voted for acceptance, two others voted for rejection. One key concern that came across in at least two reviews is that the alternative position that is being argued against is not clear: it seems to be reiterating a shared consensus that discovery requires experimentation.

Another valid concern raised by the reviewers is generalization to other scientific domains, which has not been addressed well. For instance, the paper frequently talks about ablation studies as falsification, but this would only apply to certain engineering domains. In other scientific domains, falsification explicitly has to do with scientific theories, which are a higher level of abstraction than the domains considered in this paper; yet this notion doesn't appear in the paper. The paper also does not engage with the more nuanced understanding of falsification that has now emerged in the philosophy of science.

This AC also finds the paper to be lacking in clarity (a concern raised by other reviewers as well). It is unclear  how this paper's position contrasts with the field consensus, and what it concretely means for future research directions.

Given all these concerns, this AC feels that the paper is not ready for publication.